# Encapsulate α-MnO$_2$ nanofiber within graphene layer to tune surface electronic structure for efficient ozone decomposition

Guoxiang Zhu[1,2], Wei Zhu[1,3], Yang Lou [4✉], Jun Ma[1,5], Wenqing Yao[1], Ruilong Zong[1] & Yongfa Zhu [1✉]

Major challenges encountered when developing manganese-based materials for ozone decomposition are related to the low stability and water inactivation. To solve these problems, a hierarchical structure consisted of graphene encapsulating α-MnO$_2$ nanofiber was developed. The optimized catalyst exhibited a stable ozone conversion efficiency of 80% and excellent stability over 100 h under a relative humidity (RH) of 20%. Even though the RH increased to 50%, the ozone conversion also reached 70%, well beyond the performance of α-MnO$_2$ nanofiber. Here, surface graphite carbon was activated by capturing the electron from inner unsaturated Mn atoms. The excellent stability originated from the moderate local work function, which compromised the reaction barriers in the adsorption of ozone molecule and the desorption of the intermediate oxygen species. The hydrophobic graphene shells hindered the chemisorption of water vapour, consequently enhanced its water resistance. This work offered insights for catalyst design and would promote the practical application of manganese-based catalysts in ozone decomposition.

[1] Department of Chemistry, Tsinghua University, Beijing, China. [2] Institute of Chemical Materials, China Academy of Engineering Physics, Mianyang, Sichuan, China. [3] College of Environmental and Chemical Engineering, Xi'an Polytechnic University, Xi'an, China. [4] Key Laboratory of Synthetic and Biological Colloids, Ministry of Education, School of Chemical and Material Engineering, Jiangnan University, Wuxi, Jiangsu, China. [5] College of Chemistry and Materials Science, Sichuan normal university, Chengdu, Sichuan, China. ✉email: yang.lou@jiangnan.edu.cn; zhuyf@tsinghua.edu.cn

Ground-level ozone has become one of the major air pollutants worldwide due to the massive emissions of ozone precursor, such as VOCs and NOx[1–3]. Owing to its strong oxidation potential and high reactivity, long-term exposure to even low-level of ozone would cause high morbidity in respiratory diseases[4], cardiopulmonary disease[5] and cardiovascular disease[6], especially for the elderly and children. Besides, indoor ozone would induce serious secondary organic aerosols (SOA), which is more harmful to human health than ozone itself[7]. Thus, World Health Organization requires that the maximum ozone concentration in the terms of 8 h should not exceed $100 \, \mu g \, m^{-3}$. However, severe ozone pollution frequently occurs both in developing and developed countries, especially in summer and fall. Modern household equipment involving high-pressure discharge, corona discharge or ultraviolet radiation also causes non-ignorable indoor ozone pollution, posing a severe threat to human health. Therefore, the study of ozone elimination is of great significance for environmental protection and human health.

Among the numerous ozone elimination methods[8–10], catalytic decomposition over manganese oxide, especially for $\alpha$-$MnO_2$, has attracted extensive attention due to its higher efficiency, safety and lower cost[11–14]. Based on the results of isotope labelling and in situ Raman spectroscopy, Oyama et al.[15] has proposed a mechanism of ozone decomposition over manganese oxide, in which the active site serves as electron donor and acceptor in initial ozone adsorption and final desorption of the intermediate oxygen species respectively. Our former work[16] confirms that the active site for ozone decomposition on the manganese oxide is the surface oxygen vacancy. Unfortunately, due to its small local work function, oxygen vacancy cannot capture electron easily from the intermediate oxygen species to release the active site, consequently resulting in a depressed ozone-conversion efficiency[11,17]. Besides, due to the similar chemical structure, water molecule could adsorb on the active site and hinder the adsorption of ozone molecule, which seriously decreases its performance under high humidity[16,17]. Up to now, tremendous efforts have been devoted to solving these two problems, such as transition metal doping[18], tuning the ion concentration in the tunnels of $\alpha$-$MnO_2$[19] and fabricating abundant crystal boundary[20]. However, the low stability and water-induced deactivation are still the two major challenges in ozone decomposition for manganese-based materials[16–18]. Therefore, it is highly desired to develop a new strategy to tune the surface electronic properties of manganese-based materials, aiming at accelerating its commercialisation in ozone elimination.

Graphene-based catalysts have been widely investigated because of their high stability and unique electronic properties[21]. Although pristine graphene is inert, its low density of electronic state offers the possibility to modify the electronic structure in a wide range by bringing it into contact with various materials[22]. It has been reported that the FeCo alloy encapsulated in the graphene layer enables the transfer of electrons to the graphene shell[23]. Due to the interfacial electron transfer, the work function of the encapsulated structure is between graphene and CoNi alloy, which gives moderate free energy for H* to compromise the reaction barriers of adsorption and desorption steps in hydrogen evolution reaction (HER). Graphene encapsulating Fe particles also is synthesized in their group to balance $O_2$ adsorption and $OH^-$ desorption in oxygen reduction reaction (ORR)[24]. With the help of X-ray absorption near edge structure spectra (XANES), they confirm the activity of the graphene layer originates from the electron penetration from the Fe paticles[25]. Recently, graphene-based catalysts encapsulating different metals[26,27], alloy[28–31] or metallic carbide[32] nanoparticles also have been developed to catalyse different catalytic reactions, such as ORR[24,33], HER[23], triiodide reduction reaction (IRR) in dye-sensitised solar cells (DSSCs)[30]. In addition, the smaller thickness of the graphene is, the closer the work function of the encapsulated structure is to the inner metal[34].

Inspired by these, a defective $\alpha$-$MnO_2$ nanofiber is encapsulated within ultrathin graphene shells via a one-step hydrothermal process to obtain a suitable surface electronic structure for ozone-catalytic decomposition. Experimental results confirm that the abundant oxygen vacancy is formed on the surface of inner $\alpha$-$MnO_2$ nanofiber, which donates electrons to nearby graphene shells and activates surface graphite carbon for ozone adsorption and decomposition. The transfer of interfacial electron effectively modifies the surface electronic properties, which gives the catalyst a moderate local work function to compromise the reaction barriers in the initial step of ozone adsorption and the desorption of the intermediate oxygen species. Thus, the accumulation of the intermediated oxygen species is obviously reduced, which correspondingly enhances ozone decomposition. Experimental results further confirm that the adsorption of water vapour on the active site for ozone-catalytic decomposition is weak. The high concentration of water vapour exacerbates the surface competitive adsorption and consequently results in

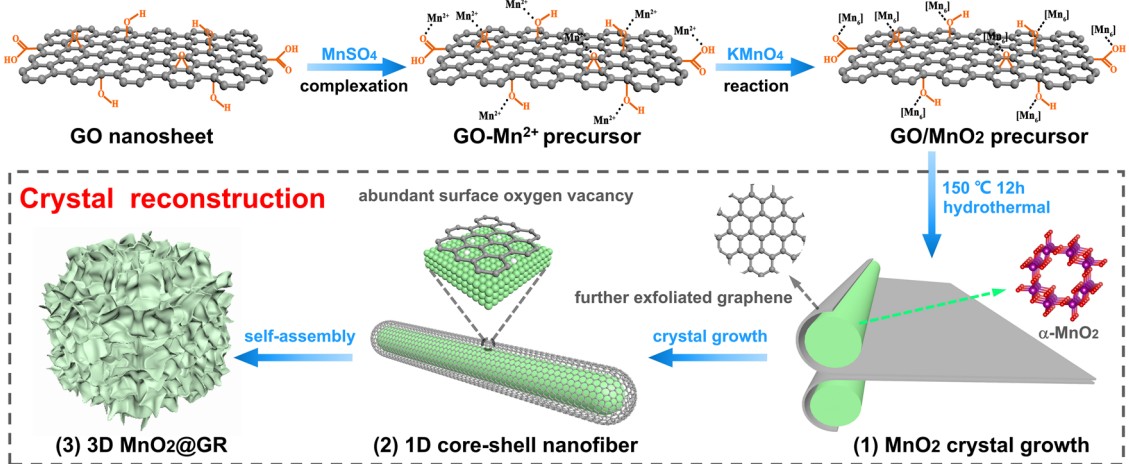

**Fig. 1 Schematic illustration of the synthetic route and model of the 3D hierarchical MnO₂@GR.** The 3D hierarchical MnO₂@GR catalysts were synthesized via the processes of the complexation of MnSO₄, oxidation of KMnO₄, the crystal growth of MnO₂ under hydrothermal condition and self-assembly of 1D core–shell nanofiber.

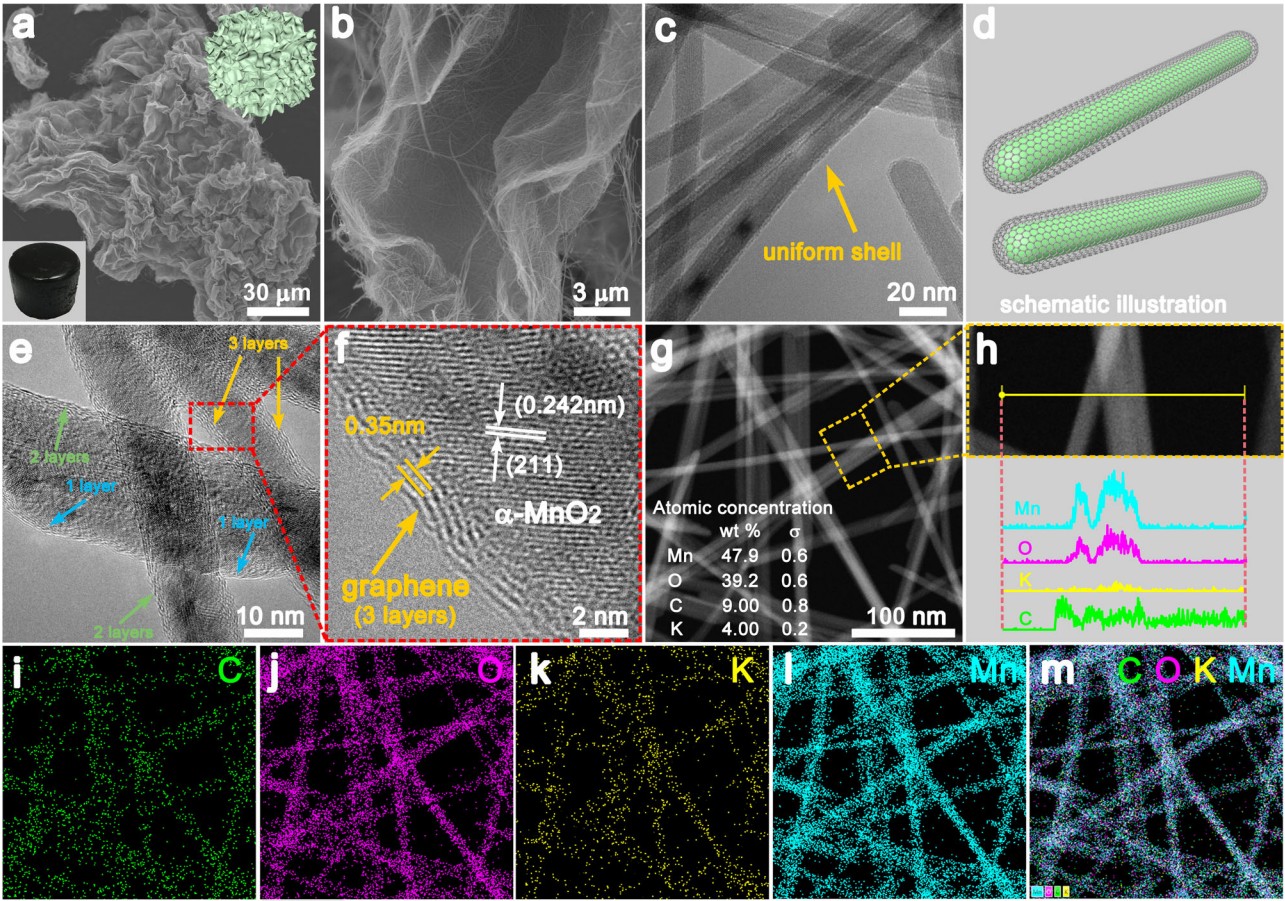

**Fig. 2 The morphology of the 3D hierarchical MnO₂@GR. a**, **b** SEM images of 7.50% MnO₂@GR. Inset: the optical images of 3D MnO₂@GR. The magnified images in (**b**) clearly reveal the 3D structure is woven from the uniform nanofiber. **c** TEM images of 7.50% MnO₂@GR, reveal the uniform core–shell structure of the nanofiber. **d** Schematic illustration of MnO₂@GR nanofiber. **e**, **f** HRTEM images of 7.50% MnO₂@GR, showing the graphene shells are about three layers (less than 2 nm). **g–m** HAADF-STEM image (**g**) and corresponding EDX linear scanning (**h**) and maps scanning of 7.50% MnO₂@GR for C (**i**), O (**j**), K (**k**), Mn (**l**) and combined image (**m**).

water-induced deactivation. In this work, the competitive adsorption of water vapour is weakened since the hydrophobic graphene shells hinder the chemisorption of water vapour. Therefore, the obtained MnO₂@GR exhibits excellent ozone-catalytic performance.

## Results

**Construction of the 3D hierarchical MnO₂@GR.** A hierarchical MnO₂@graphene (MnO₂@GR) is synthesized via one-step hydrothermal method using graphene oxide (GO) as the precursor. As illustrated in Fig. 1, α-MnO₂ nanofibers are encapsulated within graphene shells through a "complexation-reaction-growth" process. Mn²⁺ complex is formed by bonding with oxygen atoms of oxygen-containing functional groups via electrostatic force. Then, these functional groups serving as anchoring sites enable to in situ form MnO₆ octahedron units on the surface of graphene layer with the addition of KMnO₄ solution[35], which is corresponding to amorphous MnO₂ as shown in Fig. S1a, e. In the subsequent hydrothermal process, the amorphous MnO₂ is transformed into γ-MnO₂ and further into α-MnO₂ nanofiber (Fig. S1). At the same time, GO is reduced into graphene (GR) under the hydrothermal process. XRD patterns in Fig. S1e show that 7.50% MnO₂@GR possesses a pure α-MnO₂ (JCPDS No. 29-1020) crystal structure after a hydrothermal process of 12 h, indicating γ-MnO₂ is completely transformed into α-MnO₂ finally. With the crystal reconstruction of the anchored MnO₆

octahedron, the graphene layer is exfoliated to form ultrathin graphene shells. The graphene shells enable to hinder the radial growth of MnO₂, so ultralong nanofibers are obtained as shown in Fig. 2b. In addition, graphene also serves as the template for the growth of MnO₂, thus 7.50% MnO₂@GR exhibits a three-dimensional (3D) hierarchical structure (Fig. 2a–d).

**Structural characterisation of the 3D hierarchical MnO₂@GR.** Scanning electron microscopy (SEM, Fig. 2a, b) images indicate that the 3D MnO₂@GR consists of ultralong nanofibers, forming lamellar superstructure on the micrometre scale. High-resolution transmission electron microscopy (HRTEM, Fig. 2d–f) images exhibit that the ultralong nanofibers possess a uniform core–shell structure. The detailed analysis on HRTEM (Fig. 2f) image further reveals that the spacing of the lattice fringes is 0.35 nm for the shell, corresponding to the (002) plane of graphene[36], and the inner lattice fringes of 0.242 nm is attributed to the (211) plane of α-MnO₂ (JCPDS 29-1020). In addition, the graphene shells coated on the α-MnO₂ surface are very thin and most of the shells consist of only 1–3 layers (<2 nm, more details in Fig. S2). The energy-dispersive X-ray (EDS) maps (Fig. 2g, h and Fig. S3) show that the concentration of C atoms is 9.0 wt.% in 7.50% MnO₂@GR and the signal of the C atom is much stronger at the edge of the nanofiber, which further confirms the presence of graphene shells. Besides, an obviously high gloss appeared in the optical micrographs of 7.50% MnO₂@GR (Fig. S4), which further

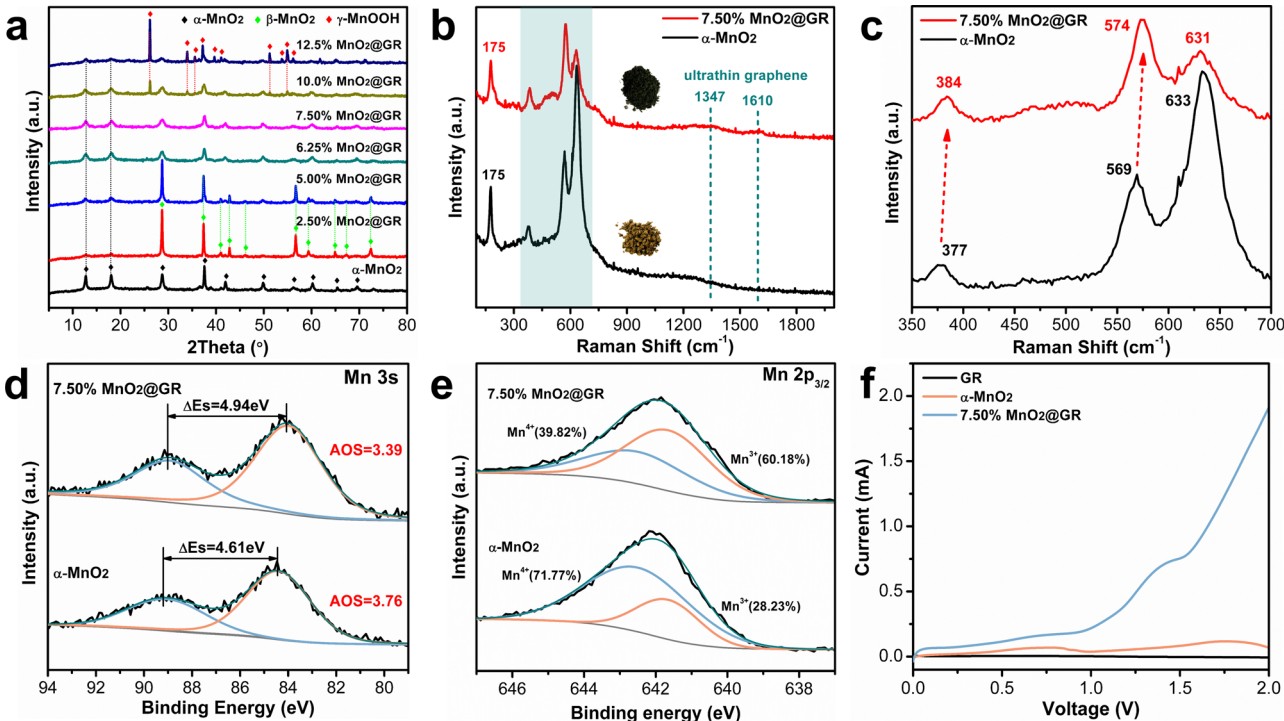

**Fig. 3 Structural analysis of the 3D hierarchical MnO₂@GR. a** XRD patterns of MnO₂@GR samples. **b** Raman shift of MnO₂ and 7.50% MnO₂@GR. Insets: optical photo of the corresponding samples. **c** Enlarged image of Raman spectra of MnO₂ and 7.50% MnO₂@GR. Mn 3s (**d**) and Mn 2p$_{3/2}$ (**e**) spectra of fresh α-MnO₂ nanowire and 7.50% MnO₂@GR. **f** CV curves of GR, MnO₂ and 7.50% MnO₂@GR in 0.1 M Bu₄NPF₆ electrolyte (0.1 M in acetonitrile).

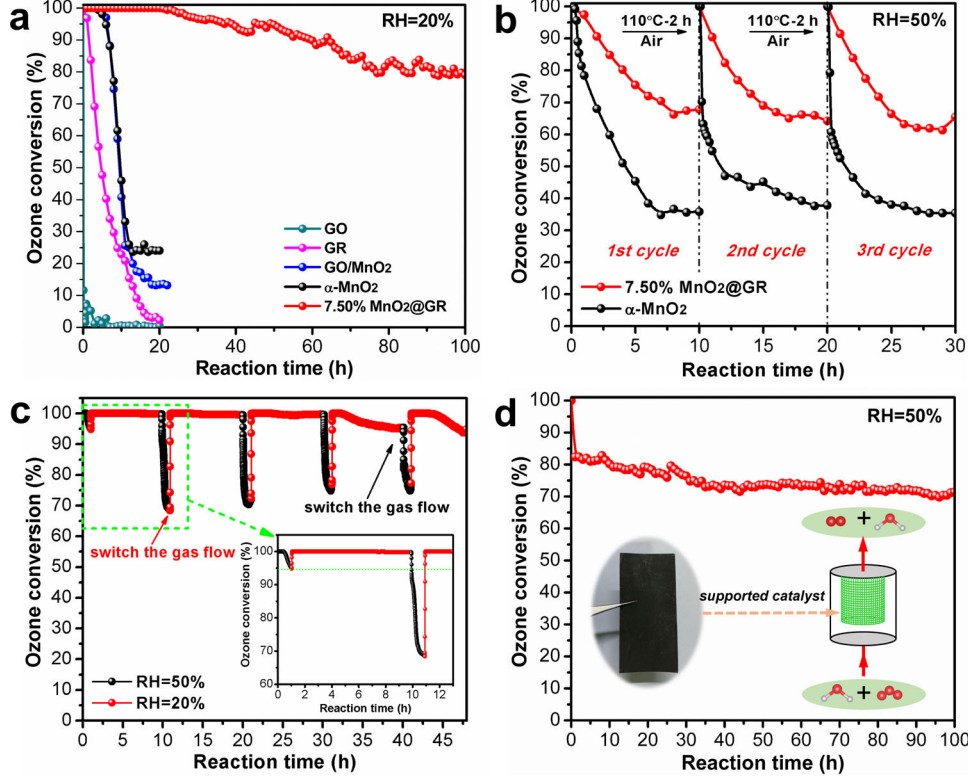

**Fig. 4 The highly efficient ozone conversion with the 3D hierarchical MnO₂@GR. a** Ozone conversion on GO, GR, α-MnO₂, GO/MnO₂ and 7.50% MnO₂@GR, respectively. **b** Ozone conversion on α-MnO₂ and 7.50% MnO₂@GR at 50%RH and their regeneration performance (Regeneration condition: 110 °C, air atmosphere). **c** Ozone conversion on 7.50% MnO₂@GR at alternate relative humidity (20% RH and 50% RH). **d** Ozone conversion of the supported catalyst at 50% RH. Inset: Photos of 7.50% MnO₂@GR coated on stainless steel mesh. Experimental conditions: 0.1 g of catalyst (0.25 g for **d**), 50 ppm O₃, flow rate = 900 mL min⁻¹, 25 °C.

confirms GR is uniformly distributed at the macroscale. These results indicate that α-MnO$_2$ nanofiber is successfully encapsulated within ultrathin graphene cages, which are further assembled into a 3D hierarchical structure.

The structures and electronic properties of MnO$_2$@GR are further investigated by X-ray diffraction (XRD), Raman spectra and X-ray photoelectron spectroscopy (XPS). XRD profiles (Fig. 3a and Fig. S5a) indicate that the crystal structure of MnO$_2$ is significantly affected by the GR contents via the strong coordination with graphene oxide. When 7.50 wt.% GO is added as a precursor, the obtained 7.50% MnO$_2$@GR possesses a pure α-MnO$_2$ (JCPDS No. 29-1020)[37] and no diffraction peaks about graphene are found. The effect of the GO contents on the morphology also is proved by SEM and TEM images and only 7.50% MnO$_2$@GR exhibits a uniform hierarchical structure (Figs. S6 and 7). Due to the presence of a large number of micropores and mesopores, the BET surface area of 7.50% MnO$_2$@GR is greatly increased to 106.7 m$^2$ g$^{-1}$ while it is 32.3 m$^2$ g$^{-1}$ for α-MnO$_2$ nanowires (Fig. S5c, d). However, the Raman spectra exhibit very weak peaks at 1347 and 1610 cm$^{-1}$, suggesting that the graphene layer is uniformly coated on α-MnO$_2$ nanofiber[38], in line with the variation of the sample colour (Fig. 3b). In the Raman spectra, the Raman shift (Fig. 3b) at 175, 384, 574, 631 cm$^{-1}$ can be attributed to α-MnO$_2$[39]. The band of 631 cm$^{-1}$ is assigned to the Mn$_3$O$_4$ formed during the process of collecting the spectrum because of the local heating of the sample[40]. As for 7.50% MnO$_2$@GR, this band is much weaker, indicating that the inner α-MnO$_2$ nanofiber is protected by the graphene shells due to its good heat conductivity. Further observation of the Raman shift (Fig. 3c), it can be found that the bands of 384 and 574 cm$^{-1}$ are slightly shifted to higher frequency compared with that of pure α-MnO$_2$ nanowires, suggesting a strong interaction between the graphene shells and inner α-MnO$_2$ nanowires[41,42]. In addition, the average oxidation state (AOS) of surface Mn atoms is estimated by the binding energy difference (ΔEs) between the two peaks of Mn 3 s and the formula used to estimate the AOS is AOS = 8.956–1.126 ΔEs[43,44]. As shown in Fig. 3d, the AOS of surface Mn atoms decreases from 3.76 to 3.39 after adding 7.5 wt.% GO as the precursor. The lower AOS of Mn atoms indicates that a large amount of low-valence Mn atoms exist on the surface of 7.50% MnO$_2$@GR. Mn 2p$_{3/2}$ peak can be deconvoluted into two peaks with the binding energy at 642.50 eV and 641.65 eV, corresponding to Mn$^{3+}$ and Mn$^{4+}$, respectively. The ratio of Mn$^{3+}$ and Mn$^{4+}$ concentration can be estimated as 1.51 in 7.50% MnO$_2$@GR based on their peak area (Fig. 3e). Oxygen vacancies will be generated to maintain electrostatic balance as long as Mn$^{3+}$ appeared in the framework of manganese dioxide, which indicates that abundant surface oxygen vacancies are formed in 7.50% MnO$_2$@GR. As shown in Fig. S8, the estimated concentration of surface oxygen species has no obvious difference after GO is added, suggesting the coated graphene layer stabilises the oxygen vacancy through their strong interaction. To further compare the concentration of the oxygen vacancy in α-MnO$_2$ and 7.50% MnO$_2$@GR, cyclic voltammetry (CV) measurement is performed in Bu$_4$NPF$_6$ (0.1 M in acetonitrile) electrolyte. As shown in Fig. 3f, the oxidation peak is negligible for GR, confirming its excellent stability in the electrolyte. For α-MnO$_2$ and 7.50% MnO$_2$@GR, the higher oxidation peak area indicates that the amount of surface oxygen vacancy is higher in 7.50% MnO$_2$@GR. These results demonstrate that the addition of GO significantly influences the crystal structure and morphology of MnO$_2$ and increases the concentration of surface oxygen vacancy.

**Excellent stability over 3D hierarchical MnO$_2$@GR.** A continuous fixed-bed reactor is used to evaluate the catalytic performance of ozone decomposition at 25 °C. As shown in Fig. 4a, the ozone conversion (20% RH) on α-MnO$_2$ nanowires starts to decrease at 3 h and drops to only 25% after 12 h, which reveals the accumulation of intermediate oxygen species on α-MnO$_2$ nanowires[19]. However, 7.50% MnO$_2$@GR exhibits 100% ozone conversion in the first 20 h and sustains the conversion rate above 80% at 100 h, showing a high and stable activity for ozone decomposition. The reference GO sample shows a negligible catalytic activity for ozone decomposition. Although the GR shows good initial catalytic activity (97%), the conversion rate dramatically drops to 0 at 20 h as shown in Fig. 4a, which might be originated from the fast consumption of nongraphitic impurities as shown in Fig. S9a. Two hybrid catalysts (GO/MnO$_2$ and GO+MnO$_2$) by physically mixing α-MnO$_2$ nanowires with the GO and GO aerogel respectively are used as the control sample to catalyse ozone decomposition. The GO/MnO$_2$ catalyst with a regular composite structure (Fig. S10) displays almost the same ozone conversion as that of α-MnO$_2$ nanowires. The GO+MnO$_2$ catalyst shows even much worse catalytic performance for ozone decomposition as shown in Fig. S9b. Therefore, it is clear that the excellent performance of 7.50% MnO$_2$@GR is attributed to the encapsulated structure even if there are some α-MnO$_2$ nanofibers that are not encapsulated by graphene.

To further explore the role of the ultrathin graphene layer in determining the catalytic performance of core–shell-structured MnO$_2$@GR, 7.50% MnO$_2$@GR is calcinated at 350 °C for 4 h under air atmosphere to remove partial graphene shells. As a result, the ozone conversion on calcinated 7.50% MnO$_2$@GR drops to 55% at 20 h as shown in Fig. S11. These results indicate that it is the unique core–shell structure rather than the graphene layer that significantly enhances the activity and stability of ozone conversion.

**Enhanced water resistance over 3D hierarchical MnO$_2$@GR.** Except for the stability, water resistance is another obstacle for the practical application of manganese-based catalysts in ozone decomposition[16,17]. Therefore, the ozone conversion is evaluated under high-humidity conditions. As shown in Fig. 4b and Fig. S12, only 35% of ozone conversion is achieved on pure α-MnO$_2$ nanowire under 50% RH, and the activity only slightly recovers after a drying process at 110 °C in air. However, for 7.50% MnO$_2$@GR, the conversion ratio can be stabilised at 67% and the activity almost recovers completely after the same drying process (Fig. 4b and Fig. S13). This indicates that the graphene shells alleviate the effect of water vapour on the ozone conversion and make catalyst regeneration much easier. When the humidity decreases from 50 to 20%, the ozone-conversion efficiency can be quickly recovered to 100% as shown in Fig. 4c, which suggests that the adsorption of H$_2$O on the 7.50% MnO$_2$@GR is weak and the regeneration can easily occur under low-humidity conditions. Those results indicate that the unique core–shell structure enables to significantly enhance the water resistance.

In order to compare the performance of the 7.50% MnO$_2$@GR with the latest reported materials (Table S1), we synthesize the OMS-2-HH[18] and MnO$_x$-HHB[20] catalysts that have been reported to possess excellent catalytic activity for ozone decomposition (Fig. S14 and S15). As shown in Fig. S16, under a relative humidity of 20%, 7.50% MnO$_2$@GR keeps 90% ozone conversion at 60 h, while OMS-2-HH and MnO$_x$-HHB catalysts only sustain 67% and 50%, respectively, under the same reaction conditions. In addition, although OMS-2-HH and MnO$_x$-HHB catalysts display higher ozone conversion under 50% RH in the first 10 h, their ozone-conversion rates decrease gradually and the deactivated catalysts could not be effectively regenerated after a drying process at 110 °C (Fig. S16b). Those experimental results

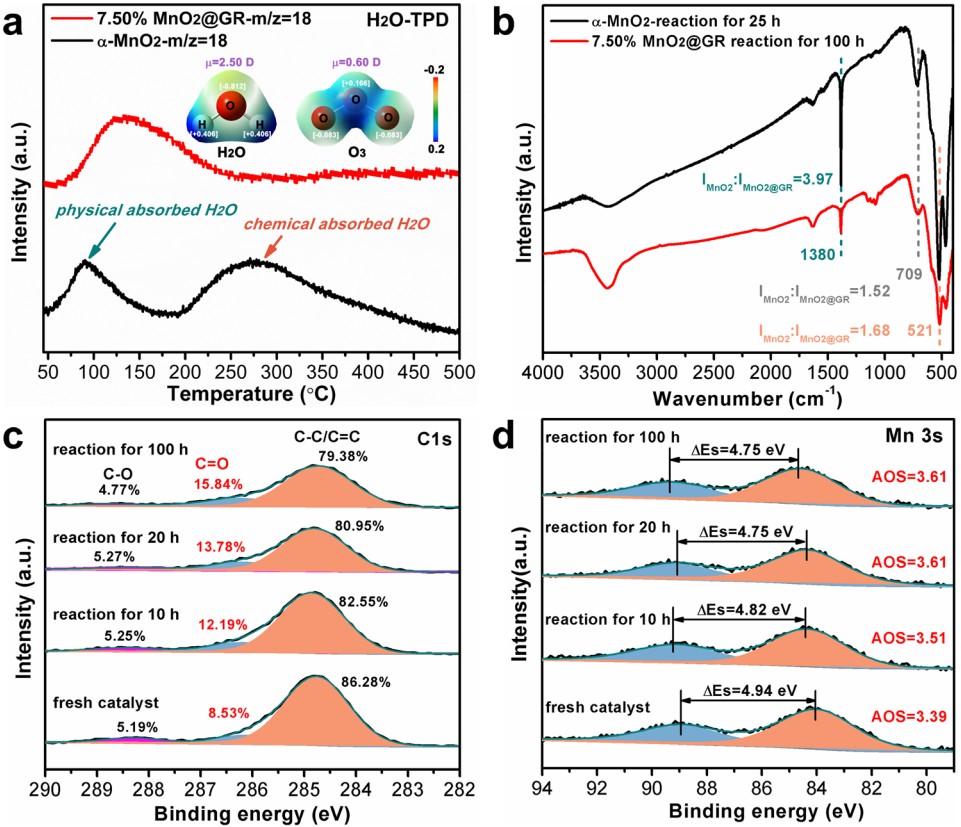

**Fig. 5 The unique advantages of MnO₂@GR in ozone conversion. a** TPD-MS profiles of MnO₂ and 7.50% MnO₂@GR. Insets: surface electrostatic potential and molecular dipole of O₃ and H₂O. **b** FTIR spectra of α-MnO₂ and 7.50% MnO₂@GR treated with O₃. Mn 3s (**c**) and C 1s (**d**) spectra of 7.50% MnO₂@GR treated with ozone for a different time.

indicate that ultrathin graphene encapsulated 7.50% MnO₂@GR possesses good water-resistance stability and regeneration ability.

Although the obtained 7.50% MnO₂@GR possesses a porous structure and could fill in the reactor directly for ozone decomposition, the gas resistance is non-negligible under a high flux and gas velocity. Therefore, loading the catalyst on a suitable support is necessary for practical application. Herein, the prepared 7.50% MnO₂@GR is uniformly coated on the wire mesh (10 × 15 cm), using the graphene layer as the framework (Fig. S17). As shown in Fig. 4d, the prepared sample exhibits a high ozone conversion of 70% and excellent stability over 100 h at a relative humidity of 50%, which indicates the potential opportunity for practical application.

**Water-resistance mechanism of MnO₂@GR.** It is generally accepted that water vapour affects the ozone conversion on manganese-based catalysts through competitive adsorption processes on the active sites[17,45,46]. However, a deep understanding on the water vapour competitive adsorption process still lacks. As shown in Fig. 4c, when the humidity increases from 20 to 50%, the ozone-conversion efficiency decreases rapidly, suggesting that the water is still adsorbed on the catalyst surface. However, the adsorbed water molecules would not affect further ozone conversion under low-humidity conditions, indicating that the active sites for water adsorption and ozone conversion are different. Therefore, it can be concluded that water vapour mainly adsorbs on the surface hydrophilic groups (such as hydroxyl groups, C=O groups and COOH groups) and affects the competitive adsorption by adsorption enrichment of the surface hydrophilic groups. To further understand the H₂O adsorption on the catalyst surface, H₂O-TPD is studied as shown in Fig. 5a.

For α-MnO₂ nanowires, water desorption appears at 90 °C and 275 °C, corresponding to the physically adsorbed water and chemically adsorbed water, respectively. However, for 7.5% MnO₂@GR, water desorption only appears at 130 °C, suggesting that the chemisorption of water vapour is hindered. These results indicate that the hydrophobic graphene shells promote the initial water resistance of 7.50% MnO₂@GR, which decreases the competitive adsorption of water vapour by hindering its chemisorption on the catalyst surface.

**Stability mechanism of MnO₂@GR.** It has been reported that ozone molecules can reversibly adsorb on the graphene and further react with graphene to form epoxide groups[47,48]. In the FTIR spectra (Fig. 5b), the peak at 1380 cm⁻¹ associated with the peroxide species can be found in 7.50% MnO₂@GR, which indicates the peroxide species are involved in the catalytic process over MnO₂@GR. A similar catalytic reaction mechanism has been reported on pure MnO₂[11]. Interestingly, the peak intensity of 7.50% MnO₂@GR reacted for 100 h is much weaker than that of α-MnO₂ nanowires reacted for 25 h as shown in Fig. 5b, suggesting that the coated graphene shells inhibit the accumulation of the peroxide species. The peak intensity associated with the peroxide species reaches the maximum intensity after 2 h (Fig. S18) but the decrease on the ozone conversion of 7.50% MnO₂@GR starts after 20 h, which suggests that the accumulation of peroxide species is no longer the main factor to limit its ozone conversion.

To explore the reason why the ozone conversion of 7.50% MnO₂@GR decreases after 20 h, the variation of the surface component is analysed via XPS data. As shown in Fig. 5c, the nongraphitic impurities in graphene shells are oxidised to C=O

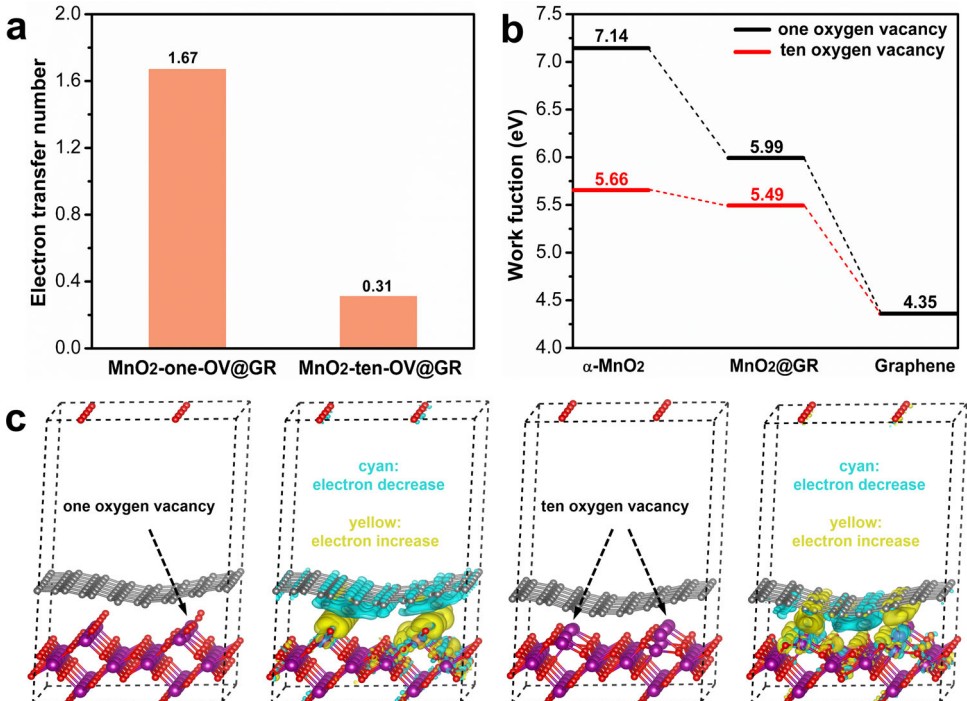

**Fig. 6 The local electronic structure in MnO₂@GR. a** The average work function of α-MnO₂, MnO₂@GR and pure graphene. **b** The electron transfer number from graphene to MnO₂ in MnO₂-one-OV@GR and MnO₂-ten-OV@GR (electron transfers from graphene layer to MnO₂). **c** The optimised structure of MnO₂@GR with different oxygen vacancy and their charge-density differences. (The yellow and cyan regions refer to the increased and decreased charge distributions, respectively. The isosurface value of the colour region is 0.001e Å⁻³. The purple, red and grey ball in the models corresponds to the Mn, oxygen and carbon atoms, respectively.).

groups and COOH groups. As shown in Fig. S19, the AOS of Mn and contents of surface adsorbed oxygen species follow the same variation trend, suggesting the oxidation state of Mn is closely related to the surface oxygen species in 7.50% MnO₂@GR even though exposed MnO₂ also can be oxidised by ozone molecule. The formation of C=O groups or COOH groups would decrease nearby electronic density, resulting in a stronger interaction between graphene shells and the inner unsaturated Mn atom. Therefore, the AOS of surface Mn atoms increases in this process (Fig. 5d). Fortunately, the ozonation process only appears on the defective structure of the graphene shells and the surface oxygen concentration would not vary with the additional ozone exposure finally (Fig. S20)[49,50]. So, although the nongraphitic impurities possess an effect on the surface electronic structure, the ozone conversion of 7.50% MnO₂@GR also keeps stable in the end.

**The origination of ozone-catalytic decomposition over MnO₂@GR.** Geunsik Lee et al.[47] has proved the dissociative chemisorption of an ozone molecule on a pure graphene layer from the physisorbed state (Fig. S21). However, the formed oxygen species (O²⁻) will not further react with the ozone molecule, suggesting its lower electron density limits the electron donation for further ozone decomposition. Therefore, it can be concluded that the surface graphene is activated in MnO₂@GR by inner α-MnO₂ nanofiber for ozone-catalytic decomposition.

To further understand the nature of ozone-catalytic decomposition over MnO₂@GR, the density functional theory simulation is conducted. The tetragonal MnO₂ with one (MnO₂-one-OV) or ten (MnO₂-ten-OV) oxygen vacancy is coated with graphene layer, respectively (Table S2) to analyse the surface electronic structure since abundant oxygen vacancies exist on the surface of 7.50% MnO₂@GR as confirmed by Mn 3 s XPS data

and CV curves (Fig. 3d–f). The optimised structures of MnO₂, graphene and MnO₂-ten-OV@GR with different graphene layers are listed in Figs. S22 and 23. Bader charge analysis (Fig. 6a) shows the electrons are transferred from the graphene layer to MnO₂, which is in line with the experimental results (Figs. S24-25). Notably, the number of transferred electrons from MnO₂ to graphene is 1.67 for MnO₂-one-OV/GR and 0.31 for MnO₂-ten-OV/GR, suggesting that the electron transfer from MnO₂ to graphene is promoted by oxygen vacancy. The average work function (Fig. 6b) also confirms this result. Second, charge-density differences (Figs. 6c and S26) reveal that electron transfer direction depends on the exposed atoms on the surface of MnO₂ in the MnO₂@GR heterojunction. For the Mn exposure site (oxygen vacancy), the electrons are transferred from Mn atoms to the nearby graphene layer and correspondingly an electron-rich site is formed, while the electrons are transferred from the graphene layer to oxygen atoms at the oxygen exposure site. The interfacial electron transfer is originated from the differences in the local work function[21]. In other words, the exposed oxygen atoms would increase the local work function of the nearby graphene layer, while the exposed Mn atoms would decrease the local work function of the nearby graphene layer. In ozone-catalytic decomposition, a lower work function is beneficial for the ozone molecule to capture electrons for further decomposition. Therefore, the electron transfer from graphite carbon to oxygen atoms reduces the surface electron density and is not beneficial for ozone-catalytic decomposition, while the electron transfer from the unsaturated Mn atoms to graphite carbon enables to increase the surface electron density and benefit the ozone-catalytic decomposition.

Since pure graphitic carbon is inert for ozone decomposition and the ozone molecule entering the interlayer of graphene sheets to react with the surface Mn species is excluded (see details in

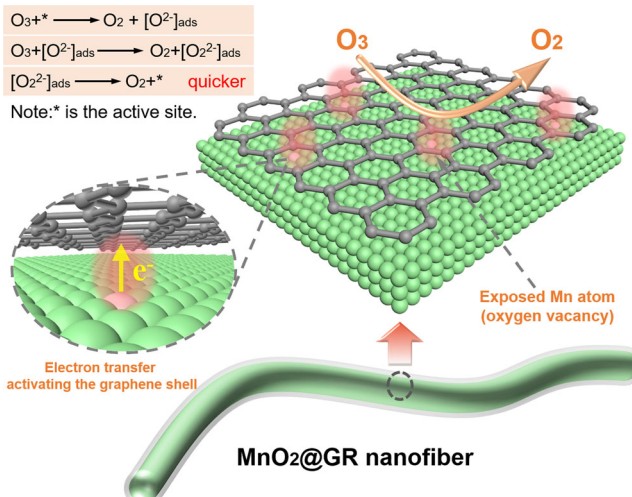

$$O_3 + * \longrightarrow O_2 + [O^{2-}]_{ads}$$
$$O_3 + [O^{2-}]_{ads} \longrightarrow O_2 + [O_2^{2-}]_{ads}$$
$$[O_2^{2-}]_{ads} \longrightarrow O_2 + * \quad \text{quicker}$$
Note:* is the active site.

Electron transfer
activating the graphene shell

Exposed Mn atom
(oxygen vacancy)

MnO₂@GR nanofiber

**Fig. 7 The schematic illustration of ozone-catalytic decomposition on MnO₂@GR.** In this scheme, the possible reaction processes of ozone-catalytic decomposition on the graphite carbon near unsaturated Mn atoms are proposed. The electron transfer from the unsaturated Mn atoms to the graphene shell turns the graphite carbon close to the unsaturated Mn atoms to be active site.

Fig. S27), we propose that the graphitic carbon close to Mn atoms (oxygen vacancy) is the active site for ozone decomposition in MnO₂@GR. The local work function of the carbon sites around oxygen vacancy is lower than that of the graphene layer and higher than that of oxygen vacancy, which compromises the reaction barriers in the initial step of ozone adsorption and the desorption of the intermediate oxygen species, consequently accelerating the desorption of peroxide species.

**Ozone-conversion mechanism over MnO₂@GR.** Ozone molecule can react with graphene to form oxygen species and peroxide species $(O_2^{2-})$ also is detected on the 7.50% MnO₂@GR. Therefore, the ozone-conversion process over MnO₂@GR is similar to that of pure α-MnO₂. Here, based on the above results and the ozone-conversion mechanism proposed in the literatures[15], the ozone decomposition mechanism on the core–shell structure of MnO₂@GR is proposed as shown in Fig. 7. The surface carbon site is activated by the electron penetration from inner unsaturated Mn atoms (oxygen vacancy). The ozone molecule is adsorbed on the activated carbon and the electron is transferred from activated carbon to the ozone molecule, leading to the formation of oxygen species $(O^{2-})$ and the release of oxygen molecule. The oxygen species $(O^{2-})$ will react with another ozone molecule to form peroxide species $(O_2^{2-})$. Finally, the peroxide species transfer one electron to the activated carbon and desorb from the active site. On the surface of MnO₂@GR, the moderate local work function compromises the reaction barriers in initial ozone adsorption and the desorption of the intermediate oxygen species, which significantly enhances the stability. The hydrophobic graphene shells inhibit the chemical adsorption of water vapour and avoid the enrichment of H₂O molecule on the catalyst surface. As a result, the 7.50% MnO₂@GR catalyst exhibits the good performance for ozone conversion.

**Conclusion.** In summary, the ultralong α-MnO₂ nanofiber is encapsulated within ultrathin graphene shells (only 1–3 layers) and further assembled into the 3D porous structure via a simple hydrothermal process. In the unique core–shell structure, the electron penetration from the oxygen vacancy of MnO₂ to nearby

graphene shells drives ozone-catalytic decomposition. Due to the interfacial charge transfer, a suitable local work function tuned by the graphene shell results in rapid decomposition of the intermediated oxygen species. Thus, 7.50% MnO₂@GR catalyst exhibits a stable ozone-conversion efficiency of 80% and excellent stability over 100 h under a relative humidity of 20%. In addition, hydrophobic graphene shells inhibit the chemical adsorption of water vapour and avoid the enrichment of H₂O molecule on the catalyst surface. So, the ozone conversion of 7.50% MnO₂@GR reaches 70% under the humidity of 50%, showing a good water resistance. These findings offer us a new perspective for the development of high-performance, stable and inexpensive catalyst and would promote manganese-based catalysts for practical application in ozone decomposition.

## Methods

**Synthesis of 3D MnO₂@GR network structure.** GO was prepared from graphite powder according to Hummer's method[51] and the concentration of the stock solution was 3.5 mg mL⁻¹. 3D MnO₂@GR was prepared through a hydrothermal process. Firstly, 1.166 g of MnSO₄·H₂O was added into 80-mL homogeneous GO aqueous dispersion (according to the mass ratio to dilute the stock solution) under continuous stirring. Subsequently, 20 mL KMnO₄ (0.727 g) solution was dropwise added into the dispersion to form a suspension with constant stir for 20 min. Then, the suspension was transformed into a 120-mL Teflon-lined stainless steel autoclave and kept at 150 °C for 12 h. After it cooled to room temperature, the products were washed with deionized (DI) water to remove the impurity. Finally, the obtained samples were directly dehydrated via a freeze-drying process. The products were marked as A% MnO₂@GR, in which A represented the mass ratio of the added GO to the theoretical yield of MnO₂.

α-MnO₂ nanowire was obtained with a similar process except without the addition of GO. For comparison, 1.0 g obtained α-MnO₂ nanowires was added into 40-mL homogeneous GO aqueous dispersion (containing 75 mg of GO) and constantly stirred for 24 h. Then, the obtained sample was directly dehydrated via a freeze-drying process and marked as GO/MnO₂. GR was obtained by a hydrothermal process. In total, 100-mL GO aqueous dispersion (containing 200 mg GO) was added into 120-mL Teflon-lined stainless steel autoclave and kept at 150 °C for 12 h. After it cooled to room temperature, the products were washed with deionized (DI) water to remove the impurity. Finally, the obtained samples were directly dehydrated via a freeze-drying process.

**Synthesis of the supported catalyst.** In all, 0.25 g of 7.50% MnO₂@GR was added into 25-mL GO aqueous dispersion (containing 50 mg of GO) under continuous stirring. Subsequently, the suspension was processed into paste at 90 °C in the presence of p-phenylenediamine. Then, the products were further coated on the wire mesh (10 × 15 cm). Finally, the sample was obtained after a heat treatment at 90 °C in a drying oven.

**Catalyst characterisation.** X-ray diffraction (XRD) patterns were collected via an X-ray diffractometer (Rigaku D/max-2400, λ = 1.5406 Å). Morphologies of the samples were obtained by a Field Emission Gun Scanning Electron Microscopy (FESEM, Hitachi SU-8010) and a transmission electron microscopy (TEM, Hitachi 7700) with an accelerating voltage of 100 kV. High-resolution transmission electron microscopy (HRTEM) images were captured via a JEM 2100F field emission transmission electron microscope at an accelerating voltage of 200 kV. The element composition and distribution were recorded by an energy-dispersive (EDS) detector equipped in JEM 2100F. XPS data was conducted in a PHI Quantera SXM™ system and the binding energy was calibrated with the signal for adventitious carbon at 284.8 eV. FTIR spectra were recorded by Bruker V70 spectrometer. CHI-660D electrochemical system was used to examine the electrochemical measurements. Electrochemical impedance spectroscopy (EIS) was measured in three-electrode quartz cells using 0.1 M Na₂SO₄ as electrolyte solution. SCE served as a reference electrode; Platinum wire served as a counter electrode, and sample film electrodes on glassy-carbon electrode served as a working electrode. Temperature programmed desorption (TPD) was carried out to on a CatLab (BEL Japan, Inc.) equipped with an online QIC-200 quadrupole mass (Inprocess Instruments, GAM 200) as a detector. The atomic force microscopy (AFM) and surface potential were obtained by Cypher VRS, Oxford, with Kelvin Probe.

**Catalyst evaluation.** The performance of catalyst for ozone decomposition was evaluated in a continuous fixed-bed reactor at 25 °C. For each test, 100 mg of catalyst was used and the gas flow rate into the reactor was maintained at 900 mL min⁻¹. Ozone was generated by arc discharge in the O₂ stream and the inlet ozone concentration was kept at 50 ± 1 ppm by tuning the discharge voltage and the gas flow rate through the Ozonator (model 1000BT-12, Shanghai Enaly Mechanical and Electrical Technology Company). Then, the generated ozone mixed adequately

with clear air in a mixing drum and then transported into the reactor. The inlet and outlet ozone concentration was recorded (model 202, 2B Technologies) and the ozone conversion was calculated through the following equation:

$$\text{Ozone removal rate} = 100\% \times (C_{in} - C_{out})/C_{in}. \qquad (1)$$

where $C_{in}$ and $C_{out}$ present inlet and outlet ozone concentration, respectively.

## Data availability
All data presented in this study are included in the article and Supplementary Information. The data are available from the corresponding authors upon request.

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

## Acknowledgements

This work was partly supported by the Chinese National Science Foundation (21437003, 21777080, 21673126, 21761142017, 21621003) and Collaborative Innovation Center for Regional Environmental Quality. UPS measurement was supported by Ms. Zhao (Zhao Zhijuan, Institute of Chemistry, Chinese Academy of Science).

## Author contributions

G.Z. synthesised the catalysts, conducted all the structural analysis and evaluated the ozone-conversion performance. W.Z. conducted the DFT calculations. J.M., W.Y. and R.Z. assisted with the material characterisations. Y.Z., Y.L. and G.Z. co-wrote the paper. Y.L. and Y.Z. supervised the work. All authors discussed the results and assisted during manuscript preparation.

## Competing interests

The authors declare no competing interests.
