## [Peer Review File · Nature Communications]

Reviewers' comments:

Reviewer #1 (Remarks to the Author):

In the manuscript "Encapsulate α -MnO₂ nanofiber within graphene layer to tune surface electronic structure for efficient ozone decomposition", Zhu et al. describe new insight that graphene layer, if encapsulated MnO₂ nanofiber, will tune the local surface electron structure and enhance the catalytic performance as well as the water resistance.

Unfortunately, the present manuscript was not suitable for publication in Nature Communications. The principal reasons are as follows:

(1) The local electron structure manipulation between graphene layer and MnO₂ in this manuscript is too limited and far from precisely controlled. This was just used for a specific reaction system, i.e., ozone decomposition, and cannot be broadened to general phenomena for gaining more fundamental insights in heterogeneous catalysis or more. It should be sent to a more discipline-specific journal.

(2) The mechanism of formation of graphene encapsulated MnO₂ nanofibers was not well stated.

(3) MnO₂@GR was mixed phases including α -MnO₂ and other underdetermined crystal phases, and γ -MnO₂ cannot be excluded from the XRD patterns in Figure S2E. DFT calculations using models based on α -MnO₂ are meaningless.

Only one layer graphene was modeled for calculation. From Figure 2e, several layers of graphene were covered on the surface of MnO₂. This will greatly affect electron transfer between MnO₂ and the surface of graphene. One layer will not support the experimental results.

(4) Element mapping confirming the presence of carbon on the surface of MnO₂ nanofibers are required. Graphene shell cannot be perfect to encapsulate MnO₂ nanofiber, and what is the junction like? The role of the junction on the catalytic performance?

(5) As the authors stated that "the graphene shells coated on the α -MnO₂ surface are very thin, and most of the shells consist of only 1~3 layers (less than 2 nm)", more evidence should be presented to confirm this.

(6) "The optical micrographs (figure S3) shown that 7.50% MnO₂@GR presented an obvious high gloss, which often appeared in graphene." Figures S3c-d are missing in Figure S3.

(7) More evidence should confirm the strong interaction between graphene and MnO₂ except for Raman. Reference is also needed to support the interaction from Raman.

(8) Primary literature rather than the cited one is encouraged for the formula: $AOS = 8.956 - 1.126\Delta Es$.

(9) It is concluded that "The lower AOS of Mn atoms means a lower average coordination number of Mn atoms and abundant oxygen vacancy on the catalyst surface". This conclusion cannot be derived from the experimental results. The encapsulation of graphene layers on the MnO₂ will decrease the number of electrons to reach MnO₂, which will increase the number of signals from the surface of MnO₂. On the contrary, more lattice signals will be collected for pure MnO₂ and the AOS also can be affected by this.

(10) Long term test of GO for ozone decomposition is needed. Ozone is one strong oxidizer and will react with GO, e.g., the surface functional groups. OMS-2-HH and MnO_x-HHB were not the best catalysts in the latest research work. This should be confirmed.

Catalytic performance reported here was not interesting for the remarkably decline (ca. 20%) under relatively low RH of 20% as well as 50%.

(11) The comparability is in doubt when 7.50% MnO₂@GR was calcined at 350 °C for 4 h. The surface structure of MnO₂ should be well characterized rather than the morphology and crystal phase. The change in the oxidation state shows great impact on the catalytic performance after calcination rather than the graphene layers.

The main reason for the decline in the catalytic performance, as shown in Figure 4a is possibly the increase in the oxidation state of Mn or the loss of oxygen vacancies (Figure 6). How did this happen if MnO₂ was well encapsulated by graphene without exposure to ozone.

(12) RH of 50% was not as high enough to confirm the water resistance of the catalyst. RH of 90% or higher is required for testing.

(13) The specific loading of 7.50% MnO₂@GR, as well as the WHSV, should be presented in Figure 4f or the results are meaningless.

(14) Are there any experimental results of the work function from UPS to support the calculated results?

(15) I notice that peaks from 500-1500 from 7.50% MnO₂@GR are all weaker than the ones from MnO₂. This was overstated. Blank tests for the samples are very important too.

(16) A Physical mixing between GO and MnO₂ is suggested. Constant stir for 24 h in aqueous conditions will destroy the surface structure of MnO₂.

(17) Other issues:

1) The title is different from the one in the manuscript

2) Line 90, corresponding must be a typo

3) Line 104, micropore and mesoporous must be micropores and mesopores

.....

The manuscript must be carefully checked!

Reviewer #2 (Remarks to the Author):

General comments:

This paper designed a catalyst (MnO₂@GR) with 3D hierarchical structure, consisted of graphene encapsulating α -MnO₂ nanofiber, and was prepared by a simple hydrothermal process. The characterization results, including SEM, TEM, XPS, Raman, etc., confirm that the MnO₂@GR with the specific 3D hierarchical structure was formed and the catalyst showed a better stability and water resistance than the unmodified α -MnO₂ nanofiber for ozone decomposition. What's more, the oxygen vacancies, which are considered as the active sites for ozone decomposition, were determined and the corresponded heterojunction models of MnO₂@GR were built with the help of DFT calculation method to confirm the synergistic effect between α -MnO₂ and GR for ozone decomposition. Besides, the reason for the high stability and water resistance of MnO₂@GR catalyst for ozone decomposition were further investigated in combination of the characterization including FT-IR, EIS, and TPD, etc. This investigation offered us a new perspective and insights on catalyst structure designation for improving the stability and water resistance of manganese oxide catalyst. So, it is acceptable for publication after a revision:

Detailed comments:

1. Page 19, L 339. In the catalyst preparation section, it is mentioned that the MnSO₄·H₂O, KMnO₄, and GO were used for the preparation of MnO₂@GR catalyst. Why MnSO₄ rather than other Mn²⁺ solution (MnNO₃, MnAc, etc.) is chosen to prepare MnO₂@GR catalysts, what is the principle of choice?

2. Page 7, L 110. The results showed that 7.50% MnO₂@GR presented a pure α -MnO₂ (JCPDS No. 29-1020) structure. Therefore, it is concluded that the cryptomelane-type α -MnO₂ nanofiber were formed based on K⁺ as tunnel template during the redox reaction between MnSO₄·H₂O, and KMnO₄. For this, ICP-OES/MS and EDS/XPS are needed to confirm the K⁺ content in the MnO₂@GR catalysts to demonstrate the validity of the proposed model.

3. The data, results and discussions, concerning about the Figure S1 and S14 (Page S4 and S19) in the SI, as well as Figure 5 (Page 16) in the manuscript, were based on the α -MnO₂ models without considering tunnel K⁺, which are considered unreasonable and the heterojunction model consisted of parallel graphene (001) and MnO₂ (110) sheet including K⁺ is more coincident with the real situation and should be rebuilt, calculated and analyzed.

4. Page 5, L 74. What's the meaning of 'intermediated oxygen vacancy'?

5. Page 9, L 144-145. The authors adopt the AOS of Mn atoms to deduce that abundant oxygen vacancies on the catalyst surface. It is not enough and the EPR measurements and XPS analysis concerning about Mn 2p_{3/2} and O 1s spectra are needed to check the active oxygen species and oxygen content on the prepared catalysts, which can further demonstrate the enhanced content of oxygen vacancies.

6. Page 16, L283-284. Why Mn 3s rather than O 1s spectra of XPS is chosen to confirm the peroxide species accumulation and desorption phenomenon.
7. Page 17, L 310-323. The ozone conversion mechanism of MnO₂@GR was proposed here. It is suggested to give the corresponded reaction mechanism diagram or chemical reaction equations for better viewing.
8. Page 17, L 310-323. As the authors and their former works have demonstrated that oxygen vacancies are the active sites for ozone catalytic decomposition (L 267-268). But the role of oxygen vacancies on ozone decomposition seems not mentioned in the ozone conversion mechanism section (Page 17, L 310-323). What's more, the promotion effect of GR for oxygen vacancies generated on MnO₂ should be clarified and summarized here.
9. Page 19, L 341-342. It is mentioned that '80 mL homogeneous GO aqueous dispersion under continuous stirring'. The company name for the production of GO should be provided and the concerned parameters such as the concentration of GO aqueous and the thickness of GO's layers etc. should be listed in the manuscript.
10. Page S12, Figure S8 d, it is mentioned that the ozone conversion of 7.50% MnO₂@GR before and after calcination. However, the activity evaluation conditions were not given and should be listed here.

Reviewer #3 (Remarks to the Author):

The authors in this MS studied alpha-MnO₂ encapsulated by graphene for O₃ decomposition at room temperature, and focused on the function of graphene in this reaction and tried to establish the correlation of the structure of the catalyst and the catalytic activities. Using various characterizations, the authors demonstrated that the graphene provided the active catalytic sites of O₃ decomposition, which could effectively make the important intermediate peroxide species desorbed via accepting two electrons from this intermediate, and that alpha-MnO₂ seem to serve as a promoter by transferring reaction electrons between the catalyst and O₃ during this reaction. Moreover, the authors also showed that the water toleration of the catalyst originated from the hydrophobic properties of the graphene. As a result, this work seems interesting, but the authors should clarify some important issues and greatly improve the quality of this MS according to the following comments, before it might be acceptable for publication.

1. It seems obscure to identify active catalytic sites in this reaction. The determination of the active site is one of the most important prerequisites for establishing reaction mechanisms. The authors concluded that the active sites were located on the graphene shells rather than encapsulated alpha-MnO₂ in lines 189-190, but did not know the accurate positions of the active sites, which were the surface -OH of graphene or surface-OH adsorbed Mn ions as shown in Figure 1 or other active carbons far away from the surface-OH of graphene. The authors reported that the nanofiber encapsulated by 2-3nm layer graphene has a uniform core-shell structure (Line 106), but in other line of 190, described as "... exposed MnO₂ nanofiber in MnO₂@GR". This indicates that the authors are not sure where the active sites were located.
2. The authors thought that the main functions of alpha-MnO₂ are to activate the graphene via their interactions, and to transfer reaction electrons during the O₃ decomposition. If this is the case, the number of alpha-MnO₂-donated electrons should equal that of alpha-MnO₂-accepted electrons for a charge balance as a reaction cycle is finished. As shown in Figure 6b, the average oxidation states of Mn increased from ~3.4 to ~3.6 after the reactions on MnO₂@GR. That indicates that some species with the negative charge like O₂⁻ or OH-species belonging to Mn also desorbed and left from MnO₂ by diffusing through the graphene to keep electrically neutral alpha-MnO₂, or this implies that the minority of MnO₂ also were exposed on the surfaces, which can reasonably provide the active sites for the O₃ decomposition.
3. The author used Mn 3s XPS to evidence that that oxygen vacant sites of MnO₂ were produced by encapsulation with graphene (Figure 3c), but this is not a solid evidence. XPS is often used as a surface tool for detecting the electronic states of the surface 2-3 nm layers. For pure alpha-MnO₂, the signal of XPS derives from the average electronic states of the Mn cations on the 2-3 nm surface layer,

but for MnO₂@GR, the signal of XPS only from the Mn cations located at no more than one nanometer surface layer due to the presence of the outmost surface 2-3 nm graphene. Since the Mn cations on the outmost surface often have a lower oxidation state than those at subsurfaces, it also appears reasonable that the detectable average oxidation states of Mn is lower after graphene covering. Thus, the authors should use other tools more sensitive to oxygen vacancy to give a more reliable result.

4. To make sure whether graphene was oxidized simultaneously during the O₃ decomposition, XPS of the fresh and used samples should be used besides citing the related references because of the different catalyst system used here from those of the references. TEM images in Figure 2 should show the graphene structures on both side-facets of one isolated alpha-MnO₂ nanofiber to evidence the successful encapsulation. Furthermore, a simple calculation should be made according to the weights of MnO₂ and graphene together with their specific surface areas to make sure whether the used graphene with the 2-3nm covering layer is enough to encapsulate 7.5% MnO₂.

5. Titles in the TEXT and the SI are different.

● **Responses to the comments of the reviewers**

Reviewer #1:

Comments:

In the manuscript “Encapsulate α -MnO₂ nanofiber within graphene layer to tune surface electronic structure for efficient ozone decomposition”, Zhu et al. describe new insight that graphene layer, if encapsulated MnO₂ nanofiber, will tune the local surface electron structure and enhance the catalytic performance as well as the water resistance.

Unfortunately, the present manuscript was not suitable for publication in Nature Communications. The principal reasons are as follows:

1. The local electron structure manipulation between graphene layer and MnO₂ in this manuscript is too limited and far from precisely controlled. This was just used for a specific reaction system, i.e., ozone decomposition, and cannot be broadened to general phenomena for gaining more fundamental insights in heterogeneous catalysis or more. It should be sent to a more discipline-specific journal.

Response: Thanks for your valuable question. As graphene is a zero-overlap semimetal, having a very low density of state at the Fermi level, the electronic properties of graphene overlayers can be tuned in a wide range by introducing different underlying metals, alloy or metallic carbide nanoparticles, thus satisfying different reaction system. This method provide us new perspective for catalyst design and surface electronic structure control. Up to now, graphene encapsulated catalyst was successfully applied in ORR, HER, triiodide reduction reaction (IRR) in dye-sensitized solar cells (DSSCs), which proved the feasibility and universality of the method. Apart from graphene layer, different 2D atomic crystal also can be used as the shell materials, such as BN, C₃N₄ and MoS₂. Therefore, this method can be introduced into different reaction system and a large amount research topics existed in the field. In this manuscript, transition metal oxides was firstly selected as the core materials to tune the electronic structure, offering a new idea for catalyst design. Besides, the encapsulated catalyst firstly applied in gas-solid reaction system, which would further expand the application scope of this type of catalyst. Therefore, this idea of catalyst design can be broadened to many reaction system. What’s more, ozone elimination is an important research topics in environmental protection and human health, so ozone catalytic

decomposition attracted more and more attention. In this work, we proposed the catalyst stability can be enhanced by modifying the local work function. Therefore, this work also was significant for the design of ozone decomposition catalyst.

2. The mechanism of formation of graphene encapsulated MnO₂ nanofibers was not well stated.

Response: Thanks for your valuable suggestion. According to your suggestion, we elaborated the formation of the graphene encapsulated MnO₂ nanofiber. The changes can be found in page 6, line 91-100 (manuscript).

3. MnO₂@GR was mixed phases including α -MnO₂ and other undertermined crystal phases, and γ -MnO₂ cannot be excluded from the XRD patterns in Figure S2E. DFT calculations using models based on α -MnO₂ are meaningless. Only one layer graphene was modeled for calculation. From Figure 2e, several layers of graphene were covered on the surface of MnO₂. This will greatly affect electron transfer between MnO₂ and the surface of graphene. One layer will not support the experimental results.

Response: Thanks for your valuable question.

Firstly, **figure S2e** shown the TEM images and XRD spectra of 7.50% MnO₂@GR with varied hydrothermal time to elaborated the crystal reconstruction process. These results indicated that the obtained 7.50% MnO₂@GR presented a pure α -MnO₂ (JCPDS No. 29-1020) after a hydrothermal process of 12 h, indicating γ -MnO₂ was completely converted to α -MnO₂ finally. In addition, HRTEM analysis (**figure 2f**) also pointed out that the lattice fringes of 0.242 nm was attributed to the (211) plane of α -MnO₂ (JCPDS 29-1020). EDS maps shown that K and Mn atoms were distributed homogeneously, suggesting that γ -MnO₂ was not existed in 7.50% MnO₂@GR because the tunnel structure was too small for K⁺. Therefore, 7.50% MnO₂@GR displayed a pure α -MnO₂ and without undetermined crystal phases.

Secondly, as the theoretical surface area of graphene was 2630 m²/g, 9.0 wt% graphene could cover 118.35 m²/g catalyst surface, almost 1.1 times BET surface area of 7.50% MnO₂@GR. Thus, the inner α -MnO₂ nanofiber would be theoretically encapsulated within 1~2 graphene layers. HRTEM images (**figure 2e-f**) also shown the graphene shells are very thin (only 1~3 layers), and

most of the graphene shells consist of only one to two layers.

Thirdly, the structure of MnO₂-ten-OV@GR with two or three graphene layers were optimized and their charge density differences also were calculated. As shown in **figure 7c** and **figure S22**, exposed Mn atoms (oxygen vacancy) would induce electron increase of the nearby graphitic carbon. **figure 7c** also shown that the amount of electron transfer decreased with the increase of the graphene layer. Thus, the local work function would vary with the graphene layer. But, unfortunately, it was hard to confirm which one was optimal for ozone catalytic decomposition.

Figure S22. The optimized structure of MnO₂-ten-OV@GR with different graphene layers and their charge density differences. (The yellow and cyan regions refer to increased and decreased charge distributions, respectively. The isosurface value of the colour region is 0.0001 e⁻Å⁻³. The purple, red and gray ball in the models corresponds to the Mn, oxygen and carbon atoms, respectively.)

4. Element mapping confirming the presence of carbon on the surface of MnO₂ nanofibers are required. Graphene shell cannot be perfect to encapsulate MnO₂ naofiber, and what is the junction like? The role of the junction on the catalytic performance?

Response: Thanks for your valuable suggestion. According to your suggestion, we added the experimental data and revised the manuscript. EDS maps and linear scanning (**figure S3**) shown no carbon signal was detected in pure α -MnO₂ nanowires. However, EDS maps (**figure 2g-h**) shown carbon content reached at 9.0 wt% in 7.50% MnO₂@GR and the signal of C atoms was stronger at the edge of the nanofiber, which confirmed the graphene encapsulated structure.

Element mapping indicated that the carbon content was 9.0 wt% in 7.50% MnO₂@GR. As the theoretical surface area of graphene was 2630 m²/g, 9.0 wt% graphene could cover 118.35

m^2/g catalyst surface, almost 1.1 times BET surface area of 7.50% $\text{MnO}_2@\text{GR}$. Thus, the inner $\alpha\text{-MnO}_2$ nanofiber would be theoretically encapsulated within 1~2 graphene layers, corresponding to the HRTEM images (**figure 2e-f**). In addition, an obvious high gloss appeared in the optical micrographs of 7.50% $\text{MnO}_2@\text{GR}$ (figure S4), indicating 7.50% $\text{MnO}_2@\text{GR}$ presented a uniform core-shell structure rather than local phenomenon. These results indicated that $\alpha\text{-MnO}_2$ nanofiber was uniformly encapsulated within ultrathin graphene cages. In the heterojunction, the local work function near the oxygen vacancy was effectively tuned by interfacial electron transfer, which compromised the reaction barriers in initial ozone adsorption and the desorption of the intermediate oxygen species, leading to a stable ozone conversion efficiency. The detail description can be found in page 14-19 (manuscript).

Figure S3. HAADF-STEM image (a) and corresponding EDX linear scanning (b) and maps scanning maps of $\alpha\text{-MnO}_2$ for K (d), O (e), Mn (f) and combined image (c).

Figure 2. The morphology of the 3D hierarchical MnO₂@GR. (a-b) SEM images of 7.50% MnO₂@GR. Inset: the optical images of 3D MnO₂@GR. The magnified images in (b) clearly reveals the 3D structure is woven from the uniform nanofiber. (c) TEM images of 7.50% MnO₂@GR, reveals the uniform core-shell structure of the nanofiber. (d) Schematic illustration of MnO₂@GR nanofiber. (e-f) HRTEM images of 7.50% MnO₂@GR, showing the graphene shells is about three layers (less than 2 nm). (g-m) HAADF-STEM image (g) and corresponding EDX linear scanning (h) and maps scanning of 7.50% MnO₂@GR for C (i), O (j), K (k), Mn (l) and combined image (m).

5. As the authors stated that “the graphene shells coated on the α-MnO₂ surface are very thin, and most of the shells consist of only 1~3 layers (less than 2 nm)”, more evidence should be presented to confirm this.

Response: Thanks for your valuable suggestion. According to your suggestion, we confirmed the carbon content firstly using EDS map scanning and 9.0 wt% carbon element existed in 7.50% MnO₂@GR. As the theoretical surface area of graphene was 2630 m²/g, 9.0 wt% graphene could cover 118.35 m²/g catalyst surface, almost 1.1 times BET surface area of 7.50% MnO₂@GR. Thus, the inner α-MnO₂ nanofiber would be theoretically encapsulated within 1~2 graphene layers, corresponding to the HRTEM images (**figure 2e-f**).

6. “The optical micrographs (figure S3) shown that 7.50% MnO₂@GR presented an obvious high gloss, which often appeared in graphene.” Figures S3c-d are missing in Figure S3.

Response: Thanks for your valuable question. We have checked the figure carefully and found the label was wrong. Now, we have corrected the mistakes and the revised figure was as following.

Figure S4. Optical micrographs of α -MnO₂ (a) and 7.50% MnO₂@GR (b) with 100 magnifications.

7. More evidence should confirm the strong interaction between graphene and MnO₂ except for Raman. Reference is also needed to support the interaction from Raman.

Response: Thanks for your valuable suggestion. Firstly, Atomic Force Microscope with a Kelvin Probe was adopted to measure the surface average potential. As shown in **figure 6a-c**, the surface potential of α -MnO₂ nanowire was higher 140 mV than the mica sheet, while that of GR was lower 80 mV than the mica sheet. For 7.50% MnO₂@GR, the surface potential was lower 15 mV than the mica sheet, between the value of α -MnO₂ nanowire and 7.50% MnO₂@GR. The difference of the surface potential indicated that electron transfer occurred on the interfacial of MnO₂@GR. As shown in **figure 6d-e**, the work function also was calculated based on the UPS data and the work function of α -MnO₂, 7.50% MnO₂@GR and GR was 4.67, 4.61 and 4.29 eV respectively. The variation of the work function also confirmed the interfacial electron transfer, suggesting a strong interaction between graphene and MnO₂. In the composite materials, spatial effect and interfacial electron transfer would its vibration energy and energy mode, resulting in a shift of the Raman band. Therefore, the shift of the Raman band also was used to evaluate the interfacial interaction, such as *Chin. J. Catal.* **41**, 302-311 (2020), *Appl Catal, B* **205**, 228-237 (2017) and *J. Am. Chem. Soc.* **136**, 5852-5855 (2014).

Figure 6. The unique surface electronic structure in $\text{MnO}_2@\text{GR}$. Surface potential of $\alpha\text{-MnO}_2$ (a), 7.50% $\text{MnO}_2@\text{GR}$ (b) and GR (c). UPS spectra of $\alpha\text{-MnO}_2$ (d), 7.50% $\text{MnO}_2@\text{GR}$ (e) and GR (f).

8. Primary literature rather than the cited one is encouraged for the formula: $\text{AOS} = 8.956 - 1.126\Delta\text{Es}$.

Response: Thanks for your valuable suggestion. The primary literature proposed the formula was *Phys. Rev. B* 65, 113102 (2002) and *Top. Catal.* 52, 470-481 (2009).

9. It is concluded that “The lower AOS of Mn atoms means a lower average coordination number of Mn atoms and abundant oxygen vacancy on the catalyst surface”. This conclusions cannot be derived from the experimental results. The encapsulation of graphene layers on the MnO_2 will decrease the number of electrons to reach MnO_2 , which will increase the number of signals from the surface of MnO_2 . On the contrary, more lattice signals will be collected for pure MnO_2 and the AOS also can be affected by this.

Response: Thanks for your valuable question. To make the statement easier to understand, we rewrite this part and added the Mn $2p_{3/2}$ spectra to confirm the results. The lower AOS of Mn atoms means a large amount of low valence Mn atoms exist on the surface of 7.50% $\text{MnO}_2@\text{GR}$. Mn $2p_{3/2}$ band also was deconvoluted into two peaks with binding energy at 642.50 eV and 641.65 eV, corresponding to Mn^{3+} and Mn^{4+} respectively. According to their peak area (**figure 3e**), it

could be found the ratio of Mn^{3+} and Mn^{4+} reached at 1.51 in 7.50% $MnO_2@GR$. As we known, once Mn^{3+} appears in the framework of manganese dioxide, oxygen vacancies will be generated to maintain electrostatic balance. Therefore, it can be concluded that abundant surface oxygen vacancy formed in 7.50% $MnO_2@GR$.

To avoid the effect of the detecting depth in XPS, cyclic voltammetry (CV) measurement was performed in Bu_4NPF_6 (0.1 M in acetonitrile) electrolyte to compare the content of the oxygen vacancy in α - MnO_2 and 7.50% $MnO_2@GR$. As shown in **figure 3f**, the oxidation peak is negligible for GR, suggesting its stability in the electrolyte. For α - MnO_2 and 7.50% $MnO_2@GR$, the oxidation peak corresponded to the oxidation of the low valence Mn atoms. The higher oxidation peak area confirmed that the amount of surface oxygen vacancy is higher in 7.50% $MnO_2@GR$.

Figure 3. Structural analysis of the 3D hierarchical $MnO_2@GR$. (a) XRD patterns of $MnO_2@GR$ samples. (b) Raman shift of MnO_2 and 7.50% $MnO_2@GR$. Insets: Optical photo of the corresponding samples. (c) Enlarged image of Raman spectra of MnO_2 and 7.50% $MnO_2@GR$. Mn 3s (d) and Mn $2p_{3/2}$ (e) spectra of fresh α - MnO_2 nanowire and 7.50% $MnO_2@GR$. (f) CV curves of GR, MnO_2 and 7.50% $MnO_2@GR$ in Bu_4NPF_6 electrolyte (0.1 M in acetonitrile).

10. Long term test of GO for ozone decomposition is needed. Ozone is one strong oxidizer and will react with GO, e.g., the surface functional groups. OMS-2-HH and MnO_x -HHB were not the best catalysts in the latest research work. This should be confirmed.

Response: Thanks for your valuable question. According to your suggestion, long term test of GO for ozone decomposition was carried out. As shown in **figure 4a**, the negligible ozone conversion of GO indicated its chemical inertness for ozone. In the Raman spectra, the peaks located at 1334 and 1597 cm^{-1} , which assigned to the G band and D band of graphene respectively and was used to quantify the density of defects in sp^2 carbon atoms. **Figure S10b** shown the integrated intensity ratio of I_D/I_G increased from 1.22 to 1.44 after a hydrothermal reduction, indicating more nongraphitic impurities formed in GR. After treated in ozone for 20 h, the ratio of I_D/I_G decreased to 1.31, suggesting the nongraphitic impurities could react with ozone molecule. Therefore, the ozone conversion over GR was not zero originally but decreased gradually with the consumption of the nongraphitic impurities. For 7.50% $\text{MnO}_2@\text{GR}$, only 9 mg carbon atoms existed in 100 mg catalyst. Without the help of the inner $\alpha\text{-MnO}_2$ nanofiber, all the carbon atoms would be consumed within 4.1 h. Therefore, it can be excluded that the excellent performance of 7.50% $\text{MnO}_2@\text{GR}$ attributed to separated graphene layer.

As the difference of the experimental conditions, it was hard to evaluate whether OMS-2-HH and $\text{MnO}_x\text{-HHB}$ were the best catalysts in the latest reported catalyst. But, OMS-2-HH and $\text{MnO}_x\text{-HHB}$ represented two typical methods to enhance the oxygen vacancy content. For OMS-2-HH, a rough surface was constructed to enhance the oxygen vacancy content. For $\text{MnO}_x\text{-HHB}$, a low crystallinity and abundant crystal boundary was fabricated to enhance the oxygen vacancy content. Besides, their catalytic performance for ozone decomposition also were good, so OMS-2-HH and $\text{MnO}_x\text{-HHB}$ were selected for comparison. To avoid misunderstanding, the statement in the manuscript was corrected.

Although 7.50% $\text{MnO}_2@\text{GR}$ experienced a decline because of the oxidation of the nongraphitic impurities, its ozone conversion still stabilized at 80 % after 100 h, suggesting a good stability for ozone conversion. After 7.50% $\text{MnO}_2@\text{GR}$ was coated on the wire mesh, its ozone conversion also kept at 70 % over 100 h under a relative humidity of 50 %. Therefore, the biggest advantages of 7.50% $\text{MnO}_2@\text{GR}$ was its stability and easy regeneration for water deactivated catalyst.

Figure 4a. Ozone conversion on GO, GR, α -MnO₂, GO/MnO₂ and 7.50% MnO₂@GR, respectively.

Figure S10b. Raman shift of GO, GR, GR-reaction (reaction for 20 h).

11. The comparability is in doubt when 7.50% MnO₂@GR was calcined at 350 °C for 4 h. The surface structure of MnO₂ should be well characterized rather than the morphology and crystal phase. The change in the oxidation state shows great impact on the catalytic performance after calcination rather than the graphene layers. The main reason for the decline in the catalytic performance, as shown in Figure 4a is possibly the increase in the

oxidation state of Mn or the loss of oxygen vacancies (Figure 6). How did this happen if MnO₂ was well encapsulated by graphene without exposure to ozone.

Response: Thanks for your valuable question. According to your suggestion, EDS map scanning was adopted to analyze the change of the carbon content after calcination. As shown in **figure S11a-b**, the calcinated process has little influence on the crystal structure and morphology, and the (EDS) maps (inset of the **figure S11b**) shown the surface graphene was almost removed. In addition, XPS was adopted to analyze the effect of calcination process on the surface structure. As shown in figure S11c-d, the ratio of the lattice oxygen and the AOS of Mn atoms increased after calcination, suggesting the surface adsorbed oxygen species transferred into lattice oxygen. Although the AOS of Mn atoms of the calcinated 7.50% MnO₂@GR reached at 3.65, the level of reacted 7.50% MnO₂@GR, its ozone conversion declined to only 55 % at 20 h, just slightly higher than the pure α -MnO₂ nanowire. However, the ozone conversion of 7.50% MnO₂@GR can kept at 80 % after 100h reaction, which has the similar AOS of Mn atoms (**figure 5c, S11c**). Therefore, these results proved again that the excellent stability attributed to the graphene encapsulated structure and the active sites for ozone decomposition was located on the graphene layer.

Figure S11. To confirm the effect of graphene layer on the ozone conversion, 7.50% MnO₂@GR was calcinated at 350 °C for 4 h under air atmosphere. (a) XRD patterns of 7.50% MnO₂@GR before and after calcination. (b) Ozone conversion of 7.50% MnO₂@GR before and after calcination and the element content for calcinated 7.50% MnO₂@GR obtained by EDS map scanning. Mn 3s (c) and O1s (d) spectra of 7.50% MnO₂@GR before and after calcination.

12. RH of 50% was not as high enough to confirm the water resistance of the catalyst. RH of 90% or higher is required for testing.

Response: Thanks for your valuable suggestion. According to your suggestion, the ozone conversion of 7.50% MnO₂@GR was evaluated under different relative humidity. As shown in **figure S16b**, the ozone conversion decreased with the increase of the relative humidity, suggesting that the competitive adsorption was still existed on 7.50% MnO₂@GR. Fortunately, only physical adsorption existed for water molecule, thus the water deactivated catalyst can be totally regenerated after a drying process at 110 °C in air.

Figure 16b. Ozone conversion on 7.50% MnO₂@GR at different relative humidity.

13. The specific loading of 7.50% MnO₂@GR, as well as the WHSV, should be presented in Figure 4f or the results are meaningless.

Response: Thanks for your valuable suggestion. The specific preparation method for supported catalyst was added in the part of catalyst preparation. The evaluation conditions and the WHSV was also added in the label of Figure 4.

Figure 4. The highly efficient ozone conversion with the 3D hierarchical $\text{MnO}_2@\text{GR}$. (a) Ozone conversion on GO, GR, $\alpha\text{-MnO}_2$, GO/MnO_2 and 7.50% $\text{MnO}_2@\text{GR}$, respectively. (b) Ozone conversion on $\alpha\text{-MnO}_2$ and 7.50% $\text{MnO}_2@\text{GR}$ at 50%RH and their regeneration performance (Regenerate condition: 110 °C, air atmosphere). (c) Ozone conversion on 7.50% $\text{MnO}_2@\text{GR}$ at alternate relative humidity (20% RH and 50% RH). (d) Ozone conversion of the supported catalyst at 50% RH. Inset: Photos of 7.50% $\text{MnO}_2@\text{GR}$ coated stainless steel mesh. Experimental conditions: 0.1 g catalyst (0.25g for (d)), 50 ppm O_3 , flow rate = 900 mL/min, 25 °C.

14. there any experimental results of the work function from UPS to support the calculated results?

Response: Thanks for your valuable suggestion. According to your suggestion, the work function was calculated based on the UPS data. As shown in **figure 6d-e**, the work function of $\alpha\text{-MnO}_2$, 7.50% $\text{MnO}_2@\text{GR}$ and GR was 4.67, 4.61 and 4.29 eV respectively. It can be found that the variation trend of the measured work function was in line with calculated results.

Figure 6. The unique surface electronic structure in MnO₂@GR. Surface potential of α -MnO₂ (a), 7.50% MnO₂@GR (b) and GR (c). UPS spectra of α -MnO₂ (d), 7.50% MnO₂@GR (e) and GR (f).

15. I notice that peaks from 500-1500 from 7.50% MnO₂@GR are all weaker than the ones from MnO₂. This was overstated. Blank tests for the samples are very important too.

Response: Thanks for your valuable question. Here, as shown in **figure 5b**, we carefully analyzed the peak intensity between 500-1500 cm⁻¹. For the peak at 1380 cm⁻¹, the ratio of the peak intensity $I_{\text{MnO}_2}/I_{\text{MnO}_2@\text{GR}}$ was 3.97. However, for the peak at 709 and 521 cm⁻¹, the ratio was 1.52 and 1.68 respectively, which indicated that the peak at 1380 cm⁻¹ was much weaker for 7.50% MnO₂@GR, comparing with that in α -MnO₂.

Figure 5b. FT-IR spectra of α -MnO₂ and 7.50% MnO₂@GR treated with O₃.

16. A Physical mixing between GO and MnO₂ is suggested. Constant stir for 24 h in aqueous conditions will destroy the surface structure of MnO₂.

Response: Thanks for your valuable suggestion. According to your suggestion, α -MnO₂ nanowires was physical mixed with 7.50 wt% GO and its performance for ozone decomposition was evaluated. As shown in **figure S10a**, the ozone conversion of GO+MnO₂ also was lower than the pure α -MnO₂ nanowires, suggesting that the excellent stability of 7.50% MnO₂@GR indeed resulted from the unique core-shell structure rather than a simple composite structure.

Figure S10a. Ozone conversion on α -MnO₂, GO/MnO₂ and GO+MnO₂ (physical mixture).

17. Other issues:

- 1) The title is different from the one in the manuscript**
- 2) Line 90, corresponding must be a typo**
- 3) Line 104, micropore and mesoporous must be micropores and mesopores**

Response: Thanks for your valuable suggestion. According to your suggestion, we have corrected the mistakes.

- 1) The title of the supporting information was corrected as “Encapsulate α -MnO₂ nanofiber within graphene layer to tune surface electronic structure for efficient ozone decomposition”.
- 2) In line 90, “conresponding” was corrected as “corresponding”.
- 3) In line 113, “micropore and mesoporous” was corrected as “micropores and mesopores”.

Reviewer #2:

Comments:

This paper designed a catalyst ($\text{MnO}_2@\text{GR}$) with 3D hierarchical structure, consisted of graphene encapsulating $\alpha\text{-MnO}_2$ nanofiber, and was prepared by a simple hydrothermal process. The characterization results, including SEM, TEM, XPS, Raman, etc., confirm that the $\text{MnO}_2@\text{GR}$ with the specific 3D hierarchical structure was formed and the catalyst showed a better stability and water resistance than the unmodified $\alpha\text{-MnO}_2$ nanofiber for ozone decomposition. What's more, the oxygen vacancies, which are considered as the active sites for ozone decomposition, were determined and the corresponded heterojunction models of $\text{MnO}_2@\text{GR}$ were built with the help of DFT calculation method to confirm the synergistic effect between $\alpha\text{-MnO}_2$ and GR for ozone decomposition. Besides, the reason for the high stability and water resistance of $\text{MnO}_2@\text{GR}$ catalyst for ozone decomposition were further investigated in combination of the characterization including FT-IR, EIS, and TPD, etc. This investigation offered us a new perspective and insights on catalyst structure designation for improving the stability and water resistance of manganese oxide catalyst. So, it is acceptable for publication after a revision:

1. Page 19, L 339. In the catalyst preparation section, it is mentioned that the $\text{MnSO}_4\cdot\text{H}_2\text{O}$, KMnO_4 , and GO were used for the preparation of $\text{MnO}_2@\text{GR}$ catalyst. Why MnSO_4 rather than other Mn^{2+} solution (MnNO_3 , MnAc , etc.) is chosen to prepare $\text{MnO}_2@\text{GR}$ catalysts, what is the principle of choice?

Response: Thanks for your valuable question. Our former work (Environ. Sci. Technol. 52, 8684-8692 (2018)) has pointed that the absorbed NO_3^- would affect the ozone conversion on the catalyst surface. Carbon dots may form in the hydrothermal process, if MnAC_2 was selected as the precursor of Mn^{2+} , which was disadvantage for the mechanism analysis in $\text{MnO}_2@\text{GR}$. Thus, MnSO_4 was selected as the precursor of Mn^{2+} .

2. Page 7, L 110. The results showed that 7.50% $\text{MnO}_2@\text{GR}$ presented a pure $\alpha\text{-MnO}_2$ (JCPDS No. 29-1020) structure. Therefore, it is concluded that the cryptomelane-type

α -MnO₂ nanofiber were formed based on K⁺ as tunnel template during the redox reaction between MnSO₄·H₂O, and KMnO₄. For this, ICP-OES/MS and EDS/XPS are needed to confirm the K⁺ content in the MnO₂@GR catalysts to demonstrate the validity of the proposed model.

Response: Thanks for your valuable question. According to your suggestion, EDS maps scanning was carried out, as shown in **figure 2g-f**, an obvious signal of K atom was detected, confirming the existence in 7.50% MnO₂@GR. The EDS maps scanning also shown that the K⁺ content was 4.00 wt%, similar with that of pure α -MnO₂ nanowires.

Figure 2. The morphology of the 3D hierarchical MnO₂@GR. (a-b) SEM images of 7.50% MnO₂@GR. Inset: the optical images of 3D MnO₂@GR. The magnified images in (b) clearly reveals the 3D structure is woven from the uniform nanofiber. (c) TEM images of 7.50% MnO₂@GR, reveals the uniform core-shell structure of the nanofiber. (d) Schematic illustration of MnO₂@GR nanofiber. (e-f) HRTEM images of 7.50% MnO₂@GR, showing the graphene shells is about three layers (less than 2 nm). (g-m) HAADF-STEM image (g) and corresponding EDX linear scanning (h) and maps scanning of 7.50% MnO₂@GR for C (i), O (j), K (k), Mn (l) and combined image (m).

3. The data, results and discussions, concerning about the Figure S1 and S14 (Page S4 and S19) in the SI, as well as Figure 5 (Page 16) in the manuscript, were based on the α -MnO₂

models without considering tunnel K^+ , which are considered unreasonable and the heterojunction model consisted of parallel graphene (001) and MnO_2 (110) sheet including K^+ is more coincident with the real situation and should be rebuilt, calculated and analyzed.

Response: Thanks for your valuable question. For α - MnO_2 , K^+ exist in the tunnels structure to stabilize the 2×2 tunnel structure. Thus, K^+ is located on the bulk rather than the catalyst surface. However, ozone catalytic decomposition occurred on the catalyst, thus inner K^+ has little effect on the surface reaction. On the other hand, the amount of the atoms in the model MnO_2 -one-OV@GR has already reached at 282. Therefore, if K^+ was further considered, the calculation time would be too long and the structure optimization would be hard to converge. So, to reduce the computations, K^+ was not considered in this work.

4. Page 5, L 74. What's the meaning of 'intermediated oxygen vacancy'?

Response: Thanks for your valuable question. We have checked this part carefully and found this is a typo. Now, "intermediated oxygen vacancy" was corrected as "intermediated oxygen species".

5. Page 9, L 144-145. The authors adopt the AOS of Mn atoms to deduce that abundant oxygen vacancies on the catalyst surface. It is not enough and the EPR measurements and XPS analysis concerning about Mn 2p_{3/2} and O 1s spectra are needed to check the active oxygen species and oxygen content on the prepared catalysts, which can further demonstrate the enhanced content of oxygen vacancies.

Response: Thanks for your valuable question. According to your suggestion, the Mn 2p_{3/2} and O 1s was measured by XPS as shown in **figure 3e** and **figure S8** respectively. Mn 2p_{3/2} band also was deconvoluted into two peaks with binding energy at 642.50 eV and 641.65 eV, corresponding to Mn^{3+} and Mn^{4+} respectively. According to their peak area (**figure 3e**), it could be found the ratio of Mn^{3+} and Mn^{4+} reached at 1.51 in 7.50% MnO_2 @GR. As we known, once Mn^{3+} appears in the framework of manganese dioxide, oxygen vacancies will be generated to maintain electrostatic balance. Therefore, it can be concluded that abundant surface oxygen vacancy formed in 7.50% MnO_2 @GR. To further compare the content of the oxygen vacancy in α - MnO_2 and 7.50% MnO_2 @GR, cyclic voltammetry (CV) measurement was performed in Bu_4NPF_6 (0.1 M in

acetonitrile) electrolyte. As shown in **figure 3f**, the oxidation peak is negligible for GR, suggesting its stability in the electrolyte. For α -MnO₂ and 7.50% MnO₂@GR, the oxidation peak corresponded to the oxidation of the low valence Mn atoms. The higher oxidation peak area indicated that the amount of surface oxygen vacancy is higher in 7.50% MnO₂@GR.

Figure 3. Structural analysis of the 3D hierarchical MnO₂@GR. (a) XRD patterns of MnO₂@GR samples. (b) Raman shift of MnO₂ and 7.50% MnO₂@GR. Insets: Optical photo of the corresponding samples. (c) Enlarged image of Raman spectra of MnO₂ and 7.50% MnO₂@GR. Mn 3s (d) and Mn 2p_{3/2} (e) spectra of fresh α -MnO₂ nanowire and 7.50% MnO₂@GR. (f) CV curves of GR, MnO₂ and 7.50% MnO₂@GR in Bu₄NPF₆ electrolyte (0.1 M in acetonitrile).

Figure S8. O 1s spectra of fresh α -MnO₂ nanowire and 7.50% MnO₂@GR.

6. Page 16, L283-284. Why Mn 3s rather than O 1s spectra of XPS is chosen to confirm the peroxide species accumulation and desorption phenomenon.

Response: Thanks for your valuable question. In fact, Mn 3s, O1s spectra of XPS are all adopted to confirm the peroxide species accumulation. As shown in figure **5c-d**, with the reaction on, the content of surface absorbed oxygen increased in the first 20 h and then kept stable, in line with the variation of the AOS of surface Mn atoms. However, FT-IF indicated that the accumulation of the intermediate oxygen species was not increased after 2 h, thus the increase of the surface absorbed oxygen resulting from the oxidation of the nongraphitic impurities. As nongraphitic impurities in graphene shells would be oxidized to C=O groups and COOH groups, corresponding to the rise of the C=O content (**figure S19**), the AOS of surface Mn atoms increased via electron transfer. Thus, it can be concluded that the decline of the ozone conversion on 7.50% MnO₂@GR after 20 h was resulting from the formation of the C=O groups and COOH groups, which changed the surface electronic structure. Fortunately, graphitic carbon was stable for ozone and the ozonation process only appeared on defect structure in the graphene shells. Thus, the ozone conversion of 7.50% MnO₂@GR also kept stable finally.

Figure 5. The unique advantages of MnO₂@GR in ozone conversion. (a) TPD-MS profiles of MnO₂ and 7.50% MnO₂@GR. Insets: Surface electrostatic potential and molecular dipole of O₃ and H₂O. (b) FT-IR spectra of α -MnO₂ and 7.50% MnO₂@GR treated with O₃. Mn 3s (c) and O 1s (d) spectra of 7.50% MnO₂@GR treated with ozone for different time.

7. Page 17, L 310-323. The ozone conversion mechanism of MnO₂@GR was proposed here. It is suggested to give the corresponded reaction mechanism diagram or chemical reaction equations for better viewing.

Response: Thanks for your valuable suggestion. According to your suggestion, a new diagram was given in **figure 8**.

Figure 8. The schematic of ozone catalytic decomposition on MnO₂@GR.

8. Page 17, L 310-323. As the authors and their former works have demonstrated that oxygen vacancies are the active sites for ozone catalytic decomposition (L 267-268). But the role of oxygen vacancies on ozone decomposition seems not mentioned in the ozone conversion mechanism section (Page 17, L 310-323). What's more, the promotion effect of GR for oxygen vacancies generated on MnO₂ should be clarified and summarized here.

Response: Thanks for your valuable suggestion. According to your suggestion, the ozone conversion mechanism over MnO₂@GR was summarized as following. Firstly, the surface carbon site was activated by the electron penetration from inner unsaturated Mn atoms (oxygen vacancy). Ozone adsorbed on the activated carbon and the electron transferred from activated carbon to ozone molecule, leading to the formation of oxygen species (O²⁻) and the release of oxygen molecule. Secondly, another ozone molecule attacked the oxygen species, forming peroxide species (O₂²⁻) and another oxygen molecule. Finally, the peroxide species gave one electron to the activated carbon and desorbed from the active site. On the surface of MnO₂@GR, the moderate local work function compromised the reaction barriers in initial ozone adsorption and the desorption of the intermediate oxygen species, leading to a stable ozone conversion efficiency. The hydrophobic graphene shells inhibited the chemical adsorption of water vapour and avoided the enrichment of H₂O molecule on the catalyst surface. Consequently, 7.50% MnO₂@GR exhibited a good performance for ozone conversion.

9. Page 19, L 341-342. It is mentioned that '80 mL homogeneous GO aqueous dispersion

under continuous stirring'. The company name for the production of GO should be provided and the concerned parameters such as the concentration of GO aqueous and the thickness of GO's layers etc. should be listed in the manuscript.

Response: Thanks for your valuable question. GO was prepared from graphite powder according to Hummer's method. The concentration of the prepared GO was 3.5 mg/mL. In this work, homogeneous GO was obtained by dilution.

10. Page S12, Figure S8 d, it is mentioned that the ozone conversion of 7.50% MnO₂@GR before and after calcination. However, the activity evaluation conditions were not given and should be listed here.

Response: Thanks for your valuable suggestion. According to your suggestion, the activity evaluation conditions were added into the label of Figure S11.

Figure S11. To confirm the effect of graphene layer on the ozone conversion, 7.50% MnO₂@GR was calcinated at 350 °C for 4 h under air atmosphere. (a) XRD patterns of 7.50% MnO₂@GR before and after calcination. (b) Ozone conversion of 7.50% MnO₂@GR before and after calcination and the element content for calcinated 7.50% MnO₂@GR obtained by EDS map scanning. Mn 3s (c) and O1s (d) spectra of 7.50%

MnO₂@GR before and after calcination. Experimental conditions: 0.1 g catalyst, 50 ppm O₃, flow rate = 900 mL/min, RH=20 %, 25 °C.

Reviewer #3:

Comments:

The authors in this MS studied alpha-MnO₂ encapsulated by graphene for O₃ decomposition at room temperature, and focused on the function of graphene in this reaction and tried to establish the correlation of the structure of the catalyst and the catalytic activities. Using various characterizations, the authors demonstrated that the graphene provided the active catalytic sites of O₃ decomposition, which could effectively make the important intermediate peroxide species desorbed via accepting two electrons from this intermediate, and that alpha-MnO₂ seem to serve as a promoter by transferring reaction electrons between the catalyst and O₃ during this reaction. Moreover, the authors also showed that the water toleration of the catalyst originated from the hydrophobic properties of the graphene. As a result, this work seems interesting, but the authors should clarify some important issues and greatly improve the quality of this MS according to the following comments, before it might be acceptable for publication.

1 . It seems obscure to identify active catalytic sites in this reaction. The determination of the active site is one of the most important prerequisites for establishing reaction mechanisms. The authors concluded that the active sites were located on the graphene shells rather than encapsulated alpha-MnO₂ in lines 189-190, but did not know the accurate positions of the active sites, which were the surface -OH of graphene or surface-OH adsorbed Mn ions as shown in Figure 1 or other active carbons far away from the surface-OH of graphene. The authors reported that the nanofiber encapsulated by 2-3nm layer graphene has a uniform core-shell structure (Line 106), but in other line of 190, described as "... exposed MnO₂ nanofiber in MnO₂@GR". This indicates that the authors are not sure where the active sites were located.

Response: Thanks for your valuable suggestion. Firstly, a series of experiments were carried out to confirm that inner α -MnO₂ nanofiber was uniformly coated. EDS maps and linear scanning (figure S3) shown no carbon signal was detected in pure α -MnO₂ nanowires. However, EDS maps (figure 2g-h) shown carbon content reached at 9.0 wt% in 7.50% MnO₂@GR and the signal

of C atoms was stronger at the edge of the nanofiber, which confirmed the graphene encapsulated structure. Element mapping indicated that the carbon content was 9.0 wt% in 7.50% MnO₂@GR. As the theoretical surface area of graphene was 2630 m²/g, 9.0 wt% graphene could cover 118.35 m²/g catalyst surface, almost 1.1 times BET surface area of 7.50% MnO₂@GR. Thus, the inner α -MnO₂ nanofiber would be theoretically encapsulated within 1~2 graphene layers, corresponding to the HRTEM images (**figure 2e-f**). In addition, an obvious high gloss appeared in the optical micrographs of 7.50% MnO₂@GR (**figure S4**), indicating 7.50% MnO₂@GR presented a uniform core-shell structure rather than local phenomenon. These results indicated that α -MnO₂ nanofiber was uniformly encapsulated within ultrathin graphene cages.

To further confirm the role of the ultrathin graphene layer, 7.50% MnO₂@GR was calcinated at 350 °C for 4 h under air atmosphere to remove partial graphene shells. As shown in **figure S11a-b**, the calcinated process has little influence on the crystal structure and morphology, and the (EDS) maps (inset of the **figure S11b**) shown the surface graphene was almost removed. XPS results (**figure S11**) shown that the ratio of the lattice oxygen and the AOS of Mn atoms increased after calcination, suggesting the surface adsorbed oxygen species transferred into lattice oxygen. Although the AOS of Mn atoms of the calcinated 7.50% MnO₂@GR reached at 3.65, the level of reacted 7.50% MnO₂@GR, its ozone conversion declined to only 55 % at 20 h, just slightly higher than the pure α -MnO₂ nanowire. However, the ozone conversion of 7.50% MnO₂@GR can kept at 80 % after 100h reaction, which has the similar AOS of Mn atoms (**figure 5c, S11c**). Therefore, these results proved again that the excellent stability attributed to the graphene encapsulated structure and the active sites for ozone decomposition was located on the graphene layer.

Finally, DFT calculation pointed out that the interfacial electron transfer direction depended on the exposed atoms on the surface of MnO₂. For the Mn exposure site (oxygen vacancy), the electrons transferred from Mn atoms to the nearby graphene layer and formed electron-rich site, while the electrons transferred from graphene layer to oxygen atoms at oxygen exposure site. The interfacial electron transfer originated from the differences in the local work function.²¹ In other words, the exposed oxygen atoms would enhance the local work function of nearby graphene layer, while the exposed Mn atoms would decrease the local work function of nearby graphene layer. In ozone catalytic decomposition, a lower work function was beneficial for the ozone molecule to

capture electrons for further decomposition. However, pure graphitic carbon was inert for ozone decomposition, so the graphitic carbon close to Mn atoms (oxygen vacancy) was the active site for ozone decomposition in MnO₂@GR.

2. The authors thought that the main functions of alpha-MnO₂ are to activate the graphene via their interactions, and to transfer reaction electrons during the O₃ decomposition. If this is the case, the number of alpha-MnO₂-donated electrons should equal that of alpha-MnO₂-accepted electrons for a charge balance as a reaction cycle is finished. As shown in Figure 6b, the average oxidation states of Mn increased from 3.4 to 3.6 after the reactions on MnO₂@GR. That indicates that some species with the negative charge like O²⁻ or OH-species belonging to Mn also desorbed and left from MnO₂ by diffusing through the graphene to keep electrically neutral alpha-MnO₂, or this implies that the minority of MnO₂ also were exposed on the surfaces, which can reasonably provide the active sites for the O₃ decomposition.

Response: Thanks for your valuable question. The ozone conversion mechanism over MnO₂@GR was illustrated in **figure 8**. Firstly, the surface carbon site was activated by the electron penetration from inner unsaturated Mn atoms (oxygen vacancy). Ozone adsorbed on the activated carbon and the electron transferred from activated carbon to ozone molecule, leading to the formation of oxygen species (O²⁻) and the release of oxygen molecule. Secondly, another ozone molecule attacked the oxygen species, forming peroxide species (O₂²⁻) and another oxygen molecule. Finally, the peroxide species gave one electron to the activated carbon and desorbed from the active site.

As shown in figure **5c-d**, with the reaction on, the content of surface absorbed oxygen increased in the first 20 h and then kept stable, in line with the variation of the AOS of surface Mn atoms. However, FT-IR indicated that the accumulation of the intermediate oxygen species was not increased after 2 h, thus the increase of the surface absorbed oxygen resulting from the oxidation of the nongraphitic impurities. As nongraphitic impurities in graphene shells would be oxidized to C=O groups and COOH groups, corresponding to the rise of the C=O content (**figure S19**), the AOS of surface Mn atoms increased via electron transfer.

Figure 8. The schematic of ozone catalytic decomposition on MnO₂@GR.

3. The author used Mn 3s XPS to evidence that that oxygen vacant sites of MnO₂ were produced by encapsulation with graphene (Figure 3c), but this is not a solid evidence. XPS is often used as a surface tool for detecting the electronic states of the surface 2-3 nm layers. For pure alpha-MnO₂, the signal of XPS derives from the average electronic states of the Mn cations on the 2-3 nm surface layer, but for MnO₂@GR, the signal of XPS only from the Mn cations located at no more than one nanometer surface layer due to the presence of the outmost surface 2-3 nm graphene. Since the Mn cations on the outmost surface often have a lower oxidation state than those at subsurfaces, it also appears reasonable that the detectable average oxidation states of Mn is lower after graphene covering. Thus, the authors should use other tools more sensitive to oxygen vacancy to give a more reliable result.

Response: Thanks for your valuable question. It indeed was difficult to compare the content of surface oxygen vacancy, as the effect of the detecting depth in XPS. Therefore, Mn 3s, Mn 2p_{3/2} just used to confirm the existence of oxygen vacancy in 7.50% MnO₂@GR. Mn 2p_{3/2} band was deconvoluted into two peaks with binding energy at 642.50 eV and 641.65 eV, corresponding to Mn³⁺ and Mn⁴⁺ respectively. According to their peak area (**figure 3e**), it could be found the ratio

of Mn^{3+} and Mn^{4+} reached at 1.51 in 7.50% $\text{MnO}_2@\text{GR}$. As we known, once Mn^{3+} appears in the framework of manganese dioxide, oxygen vacancies will be generated to maintain electrostatic balance. Therefore, it can be concluded that abundant surface oxygen vacancy formed in 7.50% $\text{MnO}_2@\text{GR}$.

To avoid the effect of the detecting depth in XPS, cyclic voltammetry (CV) measurement was performed in Bu_4NPF_6 (0.1 M in acetonitrile) electrolyte to compare the content of the oxygen vacancy in $\alpha\text{-MnO}_2$ and 7.50% $\text{MnO}_2@\text{GR}$. As shown in **figure 3f**, the oxidation peak is negligible for GR, suggesting its stability in the electrolyte. For $\alpha\text{-MnO}_2$ and 7.50% $\text{MnO}_2@\text{GR}$, the oxidation peak corresponded to the oxidation of the low valence Mn atoms. The higher oxidation peak area confirmed that the amount of surface oxygen vacancy is higher in 7.50% $\text{MnO}_2@\text{GR}$.

Figure 3. Structural analysis of the 3D hierarchical $\text{MnO}_2@\text{GR}$. (a) XRD patterns of $\text{MnO}_2@\text{GR}$ samples. (b) Raman shift of MnO_2 and 7.50% $\text{MnO}_2@\text{GR}$. Insets: Optical photo of the corresponding samples. (c) Enlarged image of Raman spectra of MnO_2 and 7.50% $\text{MnO}_2@\text{GR}$. Mn 3s (d) and Mn $2p_{3/2}$ (e) spectra of fresh $\alpha\text{-MnO}_2$ nanowire and 7.50% $\text{MnO}_2@\text{GR}$. (f) CV curves of GR, MnO_2 and 7.50% $\text{MnO}_2@\text{GR}$ in Bu_4NPF_6 electrolyte (0.1 M in acetonitrile).

4. To make sure whether graphene was oxidized simultaneously during the O3 decomposition, XPS of the fresh and used samples should be used besides citing the related references because of the different catalyst system used here from those of the references.

TEM images in Figure 2 should show the graphene structures on both side-facets of one isolated α -MnO₂ nanofiber to evidence the successful encapsulation. Furthermore, a simple calculation should be made according to the weights of MnO₂ and graphene together with their specific surface areas to make sure whether the used graphene with the 2-3nm covering layer is enough to encapsulate 7.5% MnO₂.

Response: Thanks for your valuable question. Firstly, the stability of GO, GR was evaluated. As shown in figure 4a, the negligible ozone conversion of GO indicated its chemical inertness for ozone. In the Raman spectra, the peaks located at 1334 and 1597 cm⁻¹, which assigned to the G band and D band of graphene respectively and was used to quantify the density of defects in sp² carbon atoms. Figure S10b shown the integrated intensity ratio of I_D/I_G increased from 1.22 to 1.44 after a hydrothermal reduction, indicating more nongraphitic impurities formed in GR. After treated in ozone for 20 h, the ratio of I_D/I_G decreased to 1.31, suggesting the nongraphitic impurities could react with ozone molecule. Therefore, the ozone conversion over GR was not zero originally but decreased gradually with the consumption of the nongraphitic impurities.

Figure 4a. Ozone conversion on GO, GR, α -MnO₂, GO/MnO₂ and 7.50% MnO₂@GR, respectively.

Figure S10b. Raman shift of GO, GR, GR-reaction (reaction for 20 h).

In addition, XPS spectra of the fresh and used sample also were obtained. As shown in figure **5c-d**, with the reaction on, the content of surface absorbed oxygen increased in the first 20 h and then kept stable, in line with the variation of the AOS of surface Mn atoms. However, FT-IF indicated that the accumulation of the intermediate oxygen species was not increased after 2 h, thus the increase of the surface absorbed oxygen resulting from the oxidation of the nongraphitic impurities. As nongraphitic impurities in graphene shells would be oxidized to C=O groups and COOH groups, corresponding to the rise of the C=O content (**figure S19**), the AOS of surface Mn atoms increased via electron transfer. Thus, it can be concluded that the decline of the ozone conversion on 7.50% MnO₂@GR after 20 h was resulting from the formation of the C=O groups and COOH groups, which changed the surface electronic structure. Fortunately, graphitic carbon was stable for ozone and the ozonation process only appeared on defect structure in the graphene shells. Thus, the ozone conversion of 7.50% MnO₂@GR also kept stable finally.

Figure 5. The unique advantages of MnO₂@GR in ozone conversion. (a) TPD-MS profiles of MnO₂ and 7.50% MnO₂@GR. Insets: Surface electrostatic potential and molecular dipole of O₃ and H₂O. (b) FT-IR spectra of α-MnO₂ and 7.50% MnO₂@GR treated with O₃. Mn 3s (c) and O 1s (d) spectra of 7.50% MnO₂@GR treated with ozone for different time.

To confirm whether MnO₂ nanofiber can be encapsulated completely, the carbon content was firstly measured. The energy-dispersive X-ray (EDS) maps (figure 2g-h) shown C atoms content reached at 9.0 wt% in 7.50% MnO₂@GR. As the theoretical surface area of graphene was 2630 m²/g, 9.0 wt% graphene could cover 118.35 m²/g catalyst surface, almost 1.1 times BET surface area of 7.50% MnO₂@GR. Thus, the inner α-MnO₂ nanofiber would be theoretically encapsulated within 1~2 graphene layers, in line with HRTEM images (figure 2e-f).

5. Titles in the TEXT and the SI are different.

Response: Thanks for your valuable suggestion. According to your suggestion, the title of the supporting information was corrected as “Encapsulate α-MnO₂ nanofiber within graphene layer to tune surface electronic structure for efficient ozone decomposition”.

We try our best to improve the manuscript and made some changes in the manuscript. And here we have list the changes and marked in revised paper. We appreciate for your warm work earnestly, and hope that the correction will meet with approval.

Thank you very much for consideration!

Yours Sincerely!

Prof. Yongfa Zhu

Jun. 18, 2020

REVIEWER COMMENTS

Reviewer #1 (Remarks to the Author):

The manuscript has improved while still not fulfilled the following points and cannot be published as it is:

1) The introduction of ozone decomposition as well as its relevant catalysts was too much. The scientific landscape of challenge in ozone decomposition, even in room temperature catalysis was insufficient. An intrinsic correlation between the poor water-resistant ability and the microstructure of the catalyst should be well stated, which should then inspire the design of graphene oxide encapsulated MnO₂.

The metal/metal oxide-support (graphene) interaction was the most important in modulating the catalytic performance in some cases while this was understated in the introduction section.

Overall, the introduction did not well state an interesting or universal problem as well as an expected innovative solution.

2) The authors stated that α -MnO₂ nanofiber was encapsulated by the graphene layer while only the graphene oxide layer was presented in the whole synthesis process without any reduction. Generally, the graphene layer refers to the reduced graphene oxide layer. This will mislead us and should be checked.

3) Blank space should be inserted between numbers and units, such as Figs. 2-3.

4) For ozone decomposition, typical RH=50% was commonly used while higher RH was used for the stability test, such as RH=90%. In fact, the water-resistant ability of 7.5% MnO₂@GR was not remarkable as declared. Figure 4 depicted that this catalyst deactivated rapidly at RH=50%. Only a structured catalyst overcome the deactivation, mainly involving the advantages of structured catalysts rather than the as-mentioned critical role of the graphene oxide layer on MnO₂.

7.5% MnO₂@GR was also not as active as the reported catalysts, such as MIL-100(Fe) (Angew. Chem. Int. Ed., 2018, 57(50), 16416-16420).

5) In Table S2, ozone conversion over MnO₂@GR was 70% after 100 h at RH=20% while kept 100% after 20 h or 100 h at RH=50%. I did not find the data in the manuscript.

6) Since the shell of the catalyst consist of graphene oxide layer, the surface was inevitably oxidized to various oxygen-containing groups, which were also important to change the conductivity and electron transfer of the composite and the shell. The discussion was missing in the calculation work.

7) The authors claimed that the graphitic carbon close to Mn atoms (oxygen vacancy) was the active site for ozone decomposition in MnO₂@GR. This demonstrates a confinement effect between the graphene oxide layer and MnO₂. Electron transfers from the latter one to the former one, which activated ozone toward decomposition. This is very important for identifying the active sites for ozone decomposition and for further catalyst design. In Figure 5c, the average oxidation state of Mn increased after the reaction, meaning the loss of oxygen vacancies. However, only electron transfer cannot lead to the disappearance of oxygen vacancies. Please comment on this. Does ozone diffuse from the graphene oxide layer to the surface of MnO₂?

Overall, the reported MnO₂@GR was a good rather than an outstanding catalyst for ozone decomposition considering the water-resistant ability. More importantly, the insights into the active sites only depended on the calculations without experimentally identifying the local coordination environment of the metal center. Therefore, although presented a possible method to increase the water-resistant ability, the work failed to offer an effective way for the rational design of catalysts by gaining fundamental insights into the active sites.

Reviewer #2 (Remarks to the Author):

In the revised manuscript, the K⁺ content in MnO₂@GR catalysts were confirmed by EDS scanning. CV curves and XPS analysis concerning about Mn 2p_{3/2} and O 1s spectra was added to demonstrate the enhanced content of oxygen vacancies. O 1s spectra of XPS was added to confirm the accumulation and desorption phenomenon of peroxide species. The summarized ozone conversion mechanism and the new reaction mechanism diagram make the manuscript easy to understand. The UPS data confirmed the interfacial electron transfer and supported the calculation results. The parameter of GO and the evaluation conditions also were added in the revised manuscript. Besides, some mistakes were corrected. Therefore, the revised manuscript has solved my problems and its quality was greatly improved. So, I suggested that the manuscript was accepted as it is.

Reviewer #3 (Remarks to the Author):

After the authors carefully revised and greatly improved this manuscript, now I think that it can be accepted to be published in Nature Communications. If possible, the authors can also add one extra experiment to this manuscript, which should enable it to be more reliable and more readable, because the authors claimed that the decrease in activity towards ozone decomposition originates from the nongraphitic impurities. As a result, the authors use alpha-manganese oxide encapsulated by graphene without any nongraphitic impurities, and then check the stability of the catalyst in ozone decomposition. If this is the case, such a catalyst should be stable and should be not deactivated in ozone decomposition. Of course, this may be a formidable task so that it is only an optional experiment.

- **Responses to the comments of the reviewers**

Reviewer #1:

Comments:

The manuscript has improved while still not fulfilled the following points and cannot be published as it is:

1. The introduction of ozone decomposition as well as its relevant catalysts was too much. The scientific landscape of challenge in ozone decomposition, even in room temperature catalysis was insufficient. An intrinsic correlation between the poor water-resistant ability and the microstructure of the catalyst should be well stated, which should then inspire the design of graphene oxide encapsulated MnO₂. The metal/metal oxide-support (graphene) interaction was the most important in modulating the catalytic performance in some cases while this was understated in the introduction section. Overall, the introduction did not well state an interesting or universal problem as well as an expected innovative solution.

Response: Thanks for your valuable suggestion. According to your suggestion, we have reorganized the introduction. In the new revision, the content related to ozone decomposition reduced in the introduction and large part focused on the design ideas of the article and the interfacial interaction of the core-shell structure catalyst. The detailed changes can be found in the file with track changes.

2. The authors stated that α -MnO₂ nanofiber was encapsulated by the graphene layer while only the graphene oxide layer was presented in the whole synthesis process without any reduction. Generally, the graphene layer refers to the reduced graphene oxide layer. This will mislead us and should be checked.

Response: Thanks for your valuable question. In fact, graphene oxide would be reduced in the hydrothermal process, which also was reported in *ACS Nano* 2010, 4(7), 4324-4330. Besides, **figure S10 b** shown the integrated intensity ratio of I_D/I_G increased from 1.22 to 1.44 after a hydrothermal reduction, confirming graphene oxide was indeed reduced in the hydrothermal process (*Appl. Catal., B* 2017, 205, 228-237). The former description was not clear and easy to be misunderstand. Therefore, we revised the manuscript and the changes can be found in Page 6, line

105-106.

3. Blank space should be inserted between numbers and units, such as Figs. 2-3.

Response: Thanks for your valuable suggestion. According to your suggestion, we have checked carefully and revised the manuscript.

4. For ozone decomposition, typical RH=50% was commonly used while higher RH was used for the stability test, such as RH=90%. In fact, the water-resistant ability of 7.5% MnO₂@GR was not remarkable as declared. Figure 4 depicted that this catalyst deactivated rapidly at RH=50%. Only a structured catalyst overcome the deactivation, mainly involving the advantages of structured catalysts rather than the as-mentioned critical role of the graphene oxide layer on MnO₂.

7.5% MnO₂@GR was also not as active as the reported catalysts, such as MIL-100(Fe) (Angew. Chem. Int. Ed., 2018, 57(50), 16416-16420).

Response: Thanks for your valuable question. For manganese based catalyst, the reasons for inactivation included the accumulation of the intermediate oxygen species and the competitive adsorption of water vapor. Under the low humidity, the competitive adsorption of water vapor was negligible and the deactivation mainly resulted from the accumulation of the intermediate oxygen species. Under the high humidity, the two effects were all existed. The deactivation resulted from the accumulation of the intermediate oxygen species was hard to recover while the deactivation resulted from competitive adsorption of water vapor was temporary (as shown in **figure 4c** and *Environ. Sci. Technol.* 2018, 52, 8684-8692). Therefore, RH=20 % was used to evaluate the stability and the water-resistant ability was evaluated by tuning the humidity (**figure S16b**). Considering the practical situation, the typical RH=50 % was adopted to evaluate the performance of the coated catalyst. The experimental results shown that the ozone conversion of coated catalyst can kept at 70 % at 100 h. Although the ozone conversion of 7.5% MnO₂@GR decreased with the increase of the humidity (**figure S16b**), its ozone conversion could stabilize at 67 %, largely higher than that of α -MnO₂ (35 %). More importantly, the activity of 7.5% MnO₂@GR almost recovered completely only after a drying process at 110 °C in air. Therefore, the advantages of 7.5% MnO₂@GR in water-resistant ability included the higher performance, excellent stability and good

regeneration performance.

Figure 4a has compared the performance of pure α -MnO₂ and 7.5% MnO₂@GR. The ozone conversion (20 % RH) on α -MnO₂ nanowires started to decrease at 3 h and dropped to only 25 % after 12 h. However, 7.50% MnO₂@GR exhibited 100 % ozone conversion in the first 20 h and it stabilized at 80 % even after 100 h. This results suggested that the excellent stability of 7.50% MnO₂@GR mainly depended on the unique core-shell structure. The results of AFM, USP and DFT further confirmed that the excellent stability originated from special interfacial electron transfer which gives the catalyst a moderate local work function to compromise the reaction barriers in initial ozone adsorption and the desorption of the intermediate oxygen species. Therefore, enough evidences have confirmed the critical role of the graphene layer for ozone decomposition.

The reported MIL-100(Fe) and 7.5% MnO₂@GR all have its own advantages in ozone conversion. MIL-100(Fe) (Angew. Chem. Int. Ed., 2018, 57(50), 16416-16420) indeed displayed a good performance under high humidity. However, its activity decreased with the decline of the humidity and the ozone conversion even lower than 40 % under 20 % RH. For 7.5% MnO₂@GR, the ozone conversion under low humidity was outstanding. Under typical RH=50 %, the coated catalyst also kept 70 % ozone conversion for 100 h. In addition, the deactivation resulted from water vapor could recover completely only after a drying process at 110 °C in air. At the same time, manganese based material was inexpensive and environmentally friendly. Therefore, 7.5% MnO₂@GR was more potential for commercial applications.

5. In Table S2, ozone conversion over MnO₂@GR was 70% after 100 h at RH=20% while kept 100% after 20 h or 100 h at RH=50%. I did not find the data in the manuscript.

Response: Thanks for your valuable question. We have checked the data carefully and found the data in **table S2** existed some mistakes. As shown in **figure 4a**, 7.50% MnO₂@GR exhibited 100% ozone conversion in the first 20 h and it stabilized at 80 % after 100 h under the relative humidity of 20 %. **Figure S16b** shown, under a relative humidity of 50 %, the ozone conversion of 7.50% MnO₂@GR decreased to 67 % at the first 8 h and then kept stable until 30 h. When 0.25 g 7.50% MnO₂@GR was coated on the wire mesh (10×15 cm), the ozone conversion under the relative

humidity of 50 % maintained at 70 % for 100 h (**figure 4d**). In this revision, we corrected the mistakes and the changes can be found in page S14.

6. Since the shell of the catalyst consist of graphene oxide layer, the surface was inevitably oxidized to various oxygen-containing groups, which were also important to change the conductivity and electron transfer of the composite and the shell. The discussion was missing in the calculation work.

Response: Thanks for your valuable question. Experimental data and the literatures pointed out the nongraphitic impurities in graphene shells would be oxidized to C=O groups and COOH groups. In the Raman spectra, the peaks located at 1334 and 1597 cm^{-1} , which assigned to the G band and D band of graphene respectively and was used to quantify the density of defects in sp^2 carbon atoms. **Figure S10b** shown the integrated intensity ratio of I_D/I_G increased from 1.22 to 1.44 after a hydrothermal reduction, indicating more nongraphitic impurities formed in GR. After treated in ozone for 20 h, the ratio of I_D/I_G decreased to 1.31, suggesting the nongraphitic impurities reacted with ozone molecule. Therefore, the ozone conversion over GR was not zero originally but decreased gradually with the consumption of the nongraphitic impurities. XPS results (**figure S19**) shown the nongraphitic impurities in graphene shells would be oxidized to C=O groups and COOH groups (*ACS Appl. Mater. Interfaces* 2014, 6, 1835-1842; *ACS Nano* 2011, 5, 9799-9806). The formation of the oxygen-containing functional group would decrease nearby electronic density, resulting in a stronger interaction between graphene shells and inner unsaturated Mn atom. Therefore, the AOS of surface Mn atoms increased in this process (**figure 5c**). The lower electronic density of the graphene shell would not beneficial for ozone to capture electrons. Therefore, the ozone conversion of 7.50% $\text{MnO}_2@\text{GR}$ appeared a decrease after 20 h.

Fortunately, the ozonation process only appeared on defect structure in the graphene shells and the surface oxygen concentration would not vary with additional ozone exposure finally (**figure 5d**). Therefore, the ozone conversion of 7.50% $\text{MnO}_2@\text{GR}$ also kept stable in the end.

As the formation of the oxygen-containing functional group has been proved by the experimental data, the nearby electronic density would inevitably decrease. Therefore, the changes of the surface electronic properties has already been clear. Besides, DFT calculation also only

gave out a qualitative analysis for the electron transfer. Therefore, the effect of the oxygen-containing groups was discussed based on experimental data rather than calculation. To make the manuscript more readable, we revised the manuscript and the changes can be found in Page 16, line 285-294.

7. The authors claimed that the graphitic carbon close to Mn atoms (oxygen vacancy) was the active site for ozone decomposition in MnO₂@GR. This demonstrates a confinement effect between the graphene oxide layer and MnO₂. Electron transfers from the latter one to the former one, which activated ozone toward decomposition. This is very important for identifying the active sites for ozone decomposition and for further catalyst design. In Figure 5c, the average oxidation state of Mn increased after the reaction, meaning the loss of oxygen vacancies. However, only electron transfer cannot lead to the disappearance of oxygen vacancies. Please comment on this. Does ozone diffuse from the graphene oxide layer to the surface of MnO₂?

Response: Thanks for your valuable question. The oxygen vacancy represented the Mn atoms with unsaturated coordination. Experimental data suggested that the nongraphitic impurities in graphene shells would be oxidized to C=O groups and COOH groups. The formation of the oxygen-containing functional group would decrease nearby electronic density, resulting in a stronger interaction between graphene shells and inner unsaturated Mn atom. Therefore, the AOS of surface Mn atoms increased in this process. In other words, the loss of oxygen vacancy resulted from the coordination of the nearby carbon atoms.

Reviewer #3:

Comments:

After the authors carefully revised and greatly improved this manuscript, now I think that it can be accepted to be published in Nature Communications. If possible, the authors can also add one extra experiment to this manuscript, which should enable it to be more reliable and more readable, because the authors claimed that the decrease in activity towards ozone

decomposition originates from the nongraphitic impurities. As a result, the authors use alpha-manganese oxide encapsulated by graphene without any nongraphitic impurities, and then check the stability of the catalyst in ozone decomposition. If this is the case, such a catalyst should be stable and should be not deactivated in ozone decomposition. Of course, this may be a formidable task so that it is only an optional experiment.

Response: Thanks for your valuable suggestion. According to your suggestion, we have collected a lot of literatures about the preparation of graphene. Up to now, two distinct strategies have been established for graphene synthesis: exfoliating graphite towards graphene (top-down) and building up graphene from molecular building blocks (bottom-up). The top-down methods typically include mechanical exfoliation of highly oriented pyrolyzed graphite (HOPG), solution-based exfoliation of graphite intercalation compounds (GICs), and chemical oxidation/exfoliation of graphite followed by reduction of graphene oxide (GO). The bottom-up approaches for graphene synthesis comprise epitaxial growth on metallic substrates by means of CVD, thermal decomposition of SiC, and organic synthesis based on precursor molecules. Among all of the method, mechanical exfoliation and organic synthesis are complicated and not suitable for large-scale application. Chemical oxidation/exfoliation was widely used because of its lower cost and the perspective in large-scale application. However, a large amount of nongraphitic impurities would be introduced in the chemical oxidation/exfoliation process and the oxygen-containing functional groups also were difficult to be completely reduced. In comparison, the graphene obtained by means of CVD has a higher quality. However, graphene would not form until the preparation temperature was higher than 800 °C. Under this temperature, α -MnO₂ would transfer into Mn₃O₄. Of course, nongraphitic impurities was still existed even if CVD was adopted. Therefore, up to now, it is hard to prepare α -MnO₂ encapsulated by graphene without any nongraphitic impurities.

It is a good idea to prepare α -MnO₂ encapsulated by graphene without any nongraphitic impurities for stable ozone catalytic decomposition. Although it is difficult to reach the goal under the current technical conditions, we will try more new technologies to realize it in the future. Here, thanks again to the reviewer for your valuable suggestions on this work.

We try our best to improve the manuscript and made some changes in the manuscript. All of

the changes was shown in the file with track changes. We appreciate for your warm work earnestly, and hope that the correction will meet with approval.

Thank you very much for consideration!

Yours Sincerely!

Prof. Yongfa Zhu

Sep. 16, 2020

REVIEWER COMMENTS

Reviewer #1 (Remarks to the Author):

After careful review, the article was not suggested to publish on Nature Communications under this state because the response was not convincing and some issues are still not well addressed:

(1) Materials

A well-defined catalytic material is very important to identify the real active sites. In this work, the authors claimed that the α -MnO₂ nanofiber was encapsulated by ultrathin graphene cages with 1 to 3 graphene layers.

There was only one figure (Figure 2f) displaying the microstructure of the material, in which three graphene layers were covered on the surface of α -MnO₂, rather than 1~3 layers. The conclusion that 1~3 graphene layers on the surface of MnO₂ should be addressed by MASSIVE DATA.

The authors also estimated the graphene layers to be 1~2 layers on α -MnO₂. See Figure 2f, α -MnO₂ was mainly covered by 3 graphene layers, meaning the presence of UNCOVERED SURFACE of α -MnO₂.

The experimental data in the work was not enough to support the main structure of 7.50% MnO₂@GR, i.e., graphene cage-encapsulated α -MnO₂. Exposed α -MnO₂ or exposed interface between α -MnO₂ and graphene may also exist.

Disregarding the nature of the materials will mislead the conclusions on the active sites

(2) Active sites

Surface oxygen vacancies are commonly identified by XPS O1s, ESR, O₂-TPD etc. These techniques only detected the vacancies indirectly by measuring the surface adsorbed oxygen species that possibly located on the surface oxygen vacancies. When manganese oxides were used as the catalyst, the surface oxygen species may be substituted by ozone via competitive adsorption (the mechanism is still being unknown). After preparation from hydrothermal method, the surface oxygen vacancies on manganese oxide were also possibly covered by oxygen species, even water, especially at the hydrothermal conditions. How did the surface oxygen vacancies work as the active sites just like the conventional manganese oxides.

Since the microstructure of the catalysts were not well defined for not excluding the presence of exposed MnO₂ sites as well as exposed MnO₂-graphene interface, the active sites determined herein was still controversial.

Ozone may also diffuse into the surface of graphene and interact with Mn directly. Or interact with the surface of Mn in the confined space between graphene layer and manganese oxide.

(3) Increase in the oxidation state of Mn in 7.50% MnO₂@GR

The change in the surface groups was slight, which likely not result in the remarkable change in the oxidation state of Mn. More evidence are required. The presence of exposed MnO₂ sites as well as exposed MnO₂-graphene interface will also increase the oxidation state of Mn during ozone decomposition. Or ozone can interact with MnO₂, which has been covered by graphene layer?

(4) Reaction Mechanism

The adsorption sites of water was not determined. How did water influence the competitive adsorption of ozone? Different adsorption sites only explain the superior activity of the catalyst at low RH rather than at high RH. What is the site that adsorb water molecule?

Graphitic carbon nearby surface oxygen vacancies acts as the active site toward ozone decomposition and carbon atom nearby oxygen is beneficial for the desorption of oxygen species. Since the active sites were fully different from the conventional ones over manganese oxides, the reaction mechanism may also vary. Is there any theoretical or experimental evidence for the reaction mechanism?

(5) The authors did not take a meticulous attitude towards this article. There are numerous mistakes or errors remained even after revision twice.

When a symbol consists only letters, leave a space between the number and the unit, e.g., 25 h, 5 min; when a symbol consists non-letter and letter, leave no space between the number and the unit, e.g., 25°C and 25%. The work should have been checked THOROUGHLY including the main text as well as the Figures.

There are many grammatical and spelling errors:

line 21, 'reach at' should be 'reach';

line 41, 'is of great significant' should be 'is of great significance';

line 75, 76, check the spelling of 'different';

line 121, 'reveals' should be 'reveal';

.....

(6) line 208, 209, 'the peaks located at 1334 and 1597 cm⁻¹, which assigned to the G band and D band of graphene ' should be 'the peaks located at 1334 and 1597 cm⁻¹, which are assigned to the D band and G band of graphene '

(7) 'absorb' or 'absorption' was misused in many cases (e.g., line 53, 54, 310, 313).

Reviewer #3 (Remarks to the Author):

The authors have greatly improved the quality of this manuscript after revision so that it reaches the level of Nature Communications, so I recommend it to be published in the present form.

- **Responses to the comments of the reviewers**

Reviewer #1:

Comments:

After careful review, the article was not suggested to publish on Nature Communications under this state because the response was not convincing and some issues are still not well addressed:

(1) Materials

The authors also estimated the graphene layers to be 1~2 layers on α -MnO₂. See Figure 2f, α -MnO₂ was mainly covered by 3 graphene layers, meaning the presence of uncovered surface of α -MnO₂. The experimental data in the work was not enough to support the main structure of 7.50% MnO₂@GR, i.e., graphene cage-encapsulated α -MnO₂. Exposed α -MnO₂ or exposed interface between α -MnO₂ and graphene may also exist. Disregarding the nature of the materials will mislead the conclusions on the active sites

Response: Thanks for your valuable questions. I'm sorry about the unclear description.

The calculated number of the graphene layer was based on the BET surface areas. However, the measured BET surface area was larger than the real value because of the multilayer adsorption of nitrogen molecules and large amount of the graphene edge structures. For example, the specific surface area of the typical activated carbon materials can exceed 3000 m²/g (*Nano Energy* 33 (2017) 453–461) because of the existence of the large number of meso-/microporous. On the one hand, the porous structure would result in the multilayer adsorption of nitrogen molecules, leading to a larger surface area. On the other hand, larger amount of the edge structures often existed in activated carbon, which would result in non-ignorable adsorption in the side of cyclohexane. In this manuscript, the pore size distribution (**figure S7d**) indicated the existence of the large number of meso-/microporous. Besides, the graphene layer was exfoliated during the crystal reconstruction of the anchored MnO₆ octahedron. The stress inevitably resulted in the cutting of the graphene sheet (*Angew. Chem. Int. Ed.* 2012, 51, 1161 –1164). However, the multilayer adsorption was not considered adequately during the measurement of the BET surface area. The edge structures also was not considered in the theoretical surface area of single-layer graphene. Therefore, the calculated number of the graphene layer was smaller than the real value.

As shown in **figure 2e-f**, one layer, two layers and three layers graphene are all found on the α -MnO₂ surface and more HRTEM images (**figure S5**) also confirmed this result. Besides, an obvious high gloss appeared in the optical micrographs of 7.50% MnO₂@GR (**figure S4**), further indicating 7.50% MnO₂@GR presented a uniform core-shell structure rather than local phenomenon. Therefore, these results indicated that α -MnO₂ nanofiber was successfully encapsulated within ultrathin graphene cages.

To avoid misunderstanding, the graphene structure was marked in **figure 2e** and more HRTEM images was added in **figure S5**. Besides, the manuscript also was revised and the changes can be found in page 8, line 137-141.

Figure 2. The morphology of the 3D hierarchical MnO₂@GR. (a-b) SEM images of 7.50% MnO₂@GR. Inset: the optical images of 3D MnO₂@GR. The magnified images in (b) clearly reveals the 3D structure is woven from the uniform nanofiber. (c) TEM images of 7.50% MnO₂@GR, reveal the uniform core-shell structure of the nanofiber. (d) Schematic illustration of MnO₂@GR nanofiber. (e-f) HRTEM images of 7.50% MnO₂@GR, showing the graphene shells is about three layers (less than 2 nm). (g-m) HAADF-STEM image (g) and corresponding EDX linear scanning (h) and maps scanning of 7.50% MnO₂@GR for C (i), O (j), K (k), Mn (l) and combined image (m).

Figure S5. HRTEM images of 7.50% MnO₂@GR at different regions.

(2) Active sites

Surface oxygen vacancies are commonly identified by XPS O1s, ESR, O₂-TPD etc. These techniques only detected the vacancies indirectly by measuring the surface adsorbed oxygen species that possibly located on the surface oxygen vacancies. When manganese oxides were used as the catalyst, the surface oxygen species may be substituted by ozone via competitive adsorption (the mechanism is still being unknown). After preparation from hydrothermal method, the surface oxygen vacancies on manganese oxide were also possibly covered by oxygen species, even water, especially at the hydrothermal conditions. How did the surface oxygen vacancies work as the active sites just like the conventional manganese oxides. Since the microstructure of the catalysts were not well defined for not excluding the presence of exposed MnO₂ sites as well as exposed MnO₂-graphene interface, the active sites determined herein was still controversial. Ozone may also diffuse into the surface of graphene and interact with Mn directly. Or interact with the surface of Mn in the confined space between graphene layer and manganese oxide.

Response: Thanks for your valuable question.

In this manuscript, surface oxygen vacancies were identified by XPS Mn 2p (**figure 3d**), Mn

3s (**figure 3e**) and the content of the oxygen vacancy was evaluated by electrochemical oxidation process (**figure 3f**). As we known, once Mn^{3+} appeared in the framework of manganese dioxide, oxygen vacancies will be generated to maintain electrostatic balance. Therefore, the coordination situation of Mn effectively represented the surface oxygen vacancy. Besides, For $\alpha\text{-MnO}_2$ and 7.50% $\text{MnO}_2@\text{GR}$, the oxidation peak corresponded to the oxidation of the low valence Mn atoms. The higher oxidation peak area directly indicated that the amount of surface oxygen vacancy is higher in 7.50% $\text{MnO}_2@\text{GR}$. Thus, surface oxygen vacancies was directly measured rather than indirectly by measuring the surface adsorbed oxygen species. In other words, the measured surface oxygen vacancies were not covered by oxygen species or water.

In this manuscript, contrast experiments confirmed that the outstanding performance of 7.50% $\text{MnO}_2@\text{GR}$ was attributed to the unique core-shell structure. As shown in **figure S13b**, the surface graphene was almost removed after a calcination at 350°C for 4 h under air atmosphere. As the graphene shells was destroyed, its ozone conversion at 20 h declined from 100% to only 55%. In other words, exposed MnO_2 can't exhibit such stability.

The carbon-carbon bond was 1.42Å in graphene and the covalent radius of carbon atom was 0.77Å. Thus, the biggest pore size of benzene ring was 1.30Å considering the covalent radius. For ozone molecule, the oxygen-oxygen bond was 1.28Å and its angle was 116°49' (*Catal. Rev.—Sci. Eng., 2000, 42(3), 279–322*). The covalent radius of oxygen atom was 0.66Å. Thus, the smallest size was 1.99Å. Thus, ozone molecule can't enter the confined space between graphene layer and manganese oxide by the hole of benzene ring. Although some small molecule can enter into the confined space under graphene through open channels at island edges (*Angew. Chem. 2012, 124, 4940–4943*), the process was very slow. However, ozone catalytic decomposition on 7.50% $\text{MnO}_2@\text{GR}$ was quick and high-throughput. Therefore, the surface of Mn in the confined space between graphene layer and manganese oxide was not the main active site for ozone decomposition.

Therefore, the outstanding performance of 7.50% $\text{MnO}_2@\text{GR}$ was attributed to the unique core-shell structure and the surface graphite carbon near oxygen vacancy was the active site for ozone decomposition. To make the manuscript easy to understand, some places were revised and the changes can be found in Page 13, line 234-237.

Figure S6. The model of graphene unit and ozone molecule.

(3) Increase in the oxidation state of Mn in 7.50% MnO₂@GR

The change in the surface groups was slight, which likely not result in the remarkable change in the oxidation state of Mn. More evidence are required. The presence of exposed MnO₂ sites as well as exposed MnO₂-graphene interface will also increase the oxidation state of Mn during ozone decomposition. Or ozone can interact with MnO₂, which has been covered by graphene layer?

Response: Thanks for your valuable question.

On the one hand, the formation of the surface oxygen species would induce the surface electron transfer, consequently increase the oxidation state of Mn. As shown in **figure S22**, the AOS of Mn and surface adsorbed oxygen content have same variation tendency. The variation of C=O content was slightly slow, which confirmed that the nongraphitic impurities in graphene shells would be oxidized to C=O groups and COOH groups. These results suggested the oxidation state of Mn was closely related to the surface oxygen species in 7.50% MnO₂@GR rather than directly interact with MnO₂.

On the other hand, with the oxidation of the nongraphitic impurities in graphene shells, more lattice signals of inner MnO₂ will be collected, leading to a higher AOS of Mn atoms. Because the increase of the surface oxygen species and the oxidation of the nongraphitic impurities occurred at the same time, it's hard to evaluate which one was the main reason for the increase of the AOS of Mn atoms. But, it can be ensured that the oxidation state of Mn was closely related to the surface oxygen species in 7.50% MnO₂@GR.

To make the manuscript easy to understand, some places were revised and the changes can

be found in Page 16, line 291-293.

Figure S22. The AOS of Mn, surface adsorbed oxygen content in O 1s and C=O content in C1s after 7.50% MnO₂@GR treated with ozone for different time.

(4) Reaction Mechanism

The adsorption sites of water was not determined. How did water influence the competitive adsorption of ozone? Different adsorption sites only explain the superior activity of the catalyst at low RH rather than at high RH. What is the site that adsorb water molecule?

Graphitic carbon nearby surface oxygen vacancies acts as the active site toward ozone decomposition and carbon atom nearby oxygen is beneficial for the desorption of oxygen species. Since the active sites were fully different from the conventional ones over manganese oxides, the reaction mechanism may also vary. Is there any theoretical or experimental evidence for the reaction mechanism?

Response: Thanks for your valuable question.

The adsorption sites of water on 7.50% MnO₂@GR was hard to confirm directly. Here, we proposed the point that the sites for water adsorption and ozone conversion are different based on the experimental data at alternate humidity conditions. As shown in figure 4c, the ozone conversion efficiency recovered to 100% quickly when changing the humidity from 50% to 20%, suggesting that the adsorption force was weak enough for H₂O molecule to release from the active sites and water vapour would not affect the further performance under low humidity conditions

even without regeneration. When the humidity increased again at alternate humidity conditions, the ozone conversion efficiency presented a rapid decline, suggesting that most of the absorbed water still existed on the catalyst surface after reaction in dry gas flow. If water adsorption and ozone conversion happened at the same position, the decline of the ozone conversion efficiency would be slow. On the other hand, the competitive adsorption of water vapour indeed affect ozone decomposition. Our former work also pointed out that surface hydroxide radical would exacerbate the competitive adsorption. Therefore, it was concluded that water vapour almost adsorbed on the surface hydrophilic groups (such as surface hydroxide radical) and affected the competitive adsorption by adsorption-enrichment of the surface hydrophilic groups.

The literatures reported about the interaction between ozone molecule and graphene layer. Geunsik Lee et al. (*J. Phys. Chem. C* 2009, 113, 14225–14229) investigated ozone adsorption on graphene using the ab initio density functional theory method. They pointed out that ozone molecules adsorbed on the graphene basal plane with a binding energy of 0.25 eV, and the physisorbed molecules can chemically react with graphene to form epoxide groups and oxygen molecules with a transition state of C-O-O-O. These binding energies and energy barrier indicated that the ozone adsorption on pure graphene is gentle and reversible. Except for Geunsik Lee et al., Zhiwei Xu et al. (*Chem. Eng. J.* 2014, 240, 187–194) also confirmed the C-O-C peak in the C 1s spectrum, corresponding to epoxide groups. For 7.50% MnO₂@GR, the graphitic carbon captured electrons from nearby Mn atoms (oxygen vacancies), which was more beneficial for the formation of oxygen species (O²⁻). In addition, the signal of the peroxide species (O₂²⁻) also was obtain by FTIR, as shown in **figure 5b**. Therefore, the reaction mechanism over MnO₂@GR was similar with that of pure MnO₂.

To make the manuscript easy to understand, some places were revised and the changes can be found in Page 21, line 375-380.

Figure S26. Dissociative chemisorption of an ozone molecule from the physisorbed state is shown with the transition state (*J. Phys. Chem. C* 2009, 113, 14225–14229).

(5) The authors did not take a meticulous attitude towards this article. There are numerous mistakes or errors remained even after revision twice.

When a symbol consists only letters, leave a space between the number and the unit, e.g., 25 h, 5 min; when a symbol consists non-letter and letter, leave no space between the number and the unit, e.g., 25□ and 25%. The work should have been checked thoroughly including the main text as well as the Figures.

There are many grammatical and spelling errors:

line 21, ‘reach at’ should be ‘reach’;

line 41, ‘is of great significant’ should be ‘is of great significance’;

line 75, 76, check the spelling of ‘different’;

line 121, ‘reveals’ should be ‘reveal’;

Response: Thanks for your valuable suggestions. According to your suggestions, we carefully checked the manuscript and corrected the errors. The changes can be found in the manuscript with track.

(6) line 208, 209, ‘the peaks located at 1334 and 1597 cm⁻¹, which assigned to the G band and D band of graphene’ should be ‘the peaks located at 1334 and 1597 cm⁻¹, which are assigned to the D band and G band of graphene’

Response: Thanks for your valuable suggestions. According to your suggestions, have corrected the errors and the changes can be found in

(7) ‘absorb’ or ‘absorption’ was misused in many cases (e.g., line 53, 54, 310, 313).

Response: Thanks for your valuable question. According to your suggestions, we carefully checked the manuscript and corrected the errors. The changes can be found in the manuscript with track.

We try our best to improve the manuscript and made some changes in the manuscript. All of the changes were shown in the file with track changes. We appreciate for your warm work earnestly, and hope that the correction will meet with approval.

Thank you very much for consideration!

Yours Sincerely!

Prof. Yongfa Zhu

Nov. 15, 2020

REVIEWER COMMENTS

Reviewer #1 (Remarks to the Author):

Some points were still not well addressed:

- 1) BET equation was explored for determining the surface area of mesoporous materials based on the multilayer adsorption of N₂, CO₂, Ar etc.
- 2) Even the authors showed more HRTEM images of 7.50% MnO₂@GR, the non-uniformity in graphene layers encapsulated MnO₂ and exposed MnO₂ (Fig. S5) was not convincing enough.
- 3) The coordination environment of Mn was still unclear.
- 4) The authors claimed they directly detected the presence of oxygen vacancies without oxygen species on these sites. Does this mean that the oxygen vacancies were stabilized by the covered graphene sheets?
- 5) As the graphene shells were destroyed, its ozone conversion at 20 h declined from 100% to only 55%. However, as the graphene shells were destructed, the microstructure of the encapsulated MnO₂ was also changed. The conclusion that "exposed MnO₂ can't exhibit such stability" was not supported by the experimental data.
- 6) In mean the possibility that ozone diffuse into the interlayer of graphene sheets (0.335 nm) rather than through benzene ring?
- 7) Since the adsorption sites of water was difficult to be identified, what is the effect of water molecule on the as-proposed active site?
- 8) The authors may mean the surface hydroxyl groups (-OH) rather than surface hydroxide radical (\bullet OH) for the instability of the later one on metal or metal oxide surface. Additionally, there was no effective way to detect surface hydroxide radical.

Responses to the Reviewers

Reviewer #1:

Comments: Some points were still not well addressed:

We gratefully appreciate the editor and reviewers for spending immense time on our manuscript. Our manuscript has been revised according to the valuable suggestion of reviewer. All parts revised according to the reviewer's comments are shown in the revised manuscript and supplemental information files.

Q1 BET equation was explored for determining the surface area of mesoporous materials based on the multilayer adsorption of N₂, CO₂, Ar etc.

Response: We appreciate the valuable comments of Reviewer.

In the experiment, the surface area was measured using N₂ as adsorbent, and calculated based on BET equation. By carefully checking the raw data, we found that the t-plot micropore area is 21.5 m²/g and external surface area is 85.2 m²/g, which is in line with the BET surface area (106.7 m²/g). BET equation indeed has already considered the multilayer adsorption N₂. According to the valuable suggestion of Reviewer, we have corrected the description about the BET surface area in the revised manuscript.

Q2 Even the authors showed more HRTEM images of 7.50% MnO₂@GR, the non-uniformity in graphene layers encapsulated MnO₂ and exposed MnO₂ (Fig. S5) was not convincing enough.

Response: We appreciate the valuable questions of Reviewer.

HRTEM is a powerful technology to explore the microscopic structure of nanomaterials. To our knowledge, we cannot find better method than that of HRTEM to prove the core-shell structure of synthesized 7.50% MnO₂@GR catalyst. Hence, we choose different regions on synthesized 7.50% MnO₂@GR when taking HRTEM images, trying to make a comprehensive evaluation on the core-shell structure of synthesized 7.50% MnO₂@GR catalysts. From the HRTEM images, one can find that the encapsulation layers on MnO₂ are very clear and the thickness is around 1-3 layers.

The experimental data also adequately confirm it is the unique core-shell structure of 7.50% MnO₂@GR that enhances the stability and water resistance of ozone conversion. So, the focus should be the graphene layers encapsulated MnO₂ rather than exposed MnO₂. However, we fully respect the comments of reviewer, we will try to develop new methods to quantify the extent of coverage in our following work.

Q3 The coordination environment of Mn was still unclear.

We appreciate the valuable questions of Reviewer.

In this manuscript, the synthesized 7.50% MnO₂@GR exhibits the excellent catalytic activity, stability and water-resistance for ozone decomposition. Based on our electronic and geometric characterization results and DFT simulation, we identify that the graphitic carbon close to Mn atoms (oxygen vacancy) should be the active site for ozone decomposition in MnO₂@GR. The local work function of graphene layer can be efficiently decreased by nearby Mn atoms, which is beneficial for the desorption of the intermediate oxygen species and correspondingly enhance the catalytic activity and stability for ozone decomposition.

XPS data and CV curves have confirmed the existence of the unsaturated Mn atoms. Raman spectra, UPS spectra and AFM data have verified the interfacial electron transfer. DFT simulation points out the local electron transfer occurs between surface graphite carbon and unsaturated Mn atoms.

Since the active site of synthesized 7.50% MnO₂@GR catalyst is the graphitic carbon close to Mn atoms (oxygen vacancy), the interfacial electron transfer and the local work function around the graphitic carbon are focused in the manuscript. So, we did not focus on the local coordination environment of Mn atoms. However, we fully respect the comments of reviewer, we will investigate the local coordination environment of Mn atoms in our following work.

Q4 The authors claimed they directly detected the presence of oxygen vacancies without oxygen species on these sites. Does this means that the oxygen vacancies were stabilized by the covered graphene sheets?

Response: We appreciate the valuable questions of Reviewer.

Our experimental results indicate that the ratio of Mn³⁺ and Mn⁴⁺ concentration can be estimated as 1.51 in 7.50% MnO₂@GR based on their peak area (Figure R1, Figure 3e in the main manuscript). These results indicate that the abundant surface oxygen vacancies are formed in 7.50% MnO₂@GR since oxygen vacancies will be generated to maintain electrostatic balance as long as Mn³⁺ appeared in the framework of manganese dioxide.

The average oxidation state (AOS) of surface Mn atoms represents the concentration of oxygen vacancies to some extent, which is estimated by the binding energy difference (ΔE_s) between the two peaks of Mn 3s with a formula of $AOS = 8.956 - 1.126\Delta E_s$ (Top. Catal. 2009, 52, 470-481; Phys. Rev. B 2002, 65, 113102). The AOS of surface Mn species in α -MnO₂ nanowires is 3.76 eV while it is only 3.39 eV in 7.50% MnO₂@GR, which indicates that there are more oxygen vacancies present on the surface of synthesized 7.50% MnO₂@GR. Those results indicate that the presence of graphene layers encapsulating on α -MnO₂ nanowires benefits to form more oxygen vacancies.

XPS data of O 1s indicates the concentration of surface oxygen species has no obvious difference between 7.50% MnO₂@GR and α -MnO₂, suggesting the oxygen vacancies are not stabilized by surface oxygen species. Raman spectra, UPS spectra and AFM data have verified the interfacial electron transfer. DFT simulation points out the local electron transfer occurs between surface graphite carbon and unsaturated Mn atoms. These results

suggest that the formed oxygen vacancies are stabilized by the strong interaction between 7.50% MnO₂@GR and α -MnO₂.

Figure R1. Spectra of fresh α -MnO₂ nanowire and 7.50% MnO₂@GR.

Q5 As the graphene shells was destroyed, its ozone conversion at 20 h declined from 100% to only 55%. However, as the graphene shells was destructed, the microstructure of the encapsulated MnO₂ was also changed. The conclusion that “exposed MnO₂ can’t exhibit such stability” was not supported by the experimental data.

Response: We appreciate the valuable comments of Reviewer.

Our stability data confirms that the ozone conversion gradually decreases from 100% to only 55% at 20 h when the graphene shells are destroyed, which indicates that the unique core-shell structure enables to significantly enhance the activity and stability of ozone conversion. Since there is a strong interaction between MnO₂ and graphene shells, we agree with the comment of reviewer that the microstructure of the encapsulated MnO₂ may be changed when the graphene shells is destructed. Therefore, we prepared pure α -MnO₂ as reference. If exposed MnO₂ existed in 7.50% MnO₂@GR, the microstructure of exposed MnO₂ would be same with pure α -MnO₂. However, as shown in Figure 4a, the ozone conversion (20% RH) on α -MnO₂ nanowires started to decrease at 3 h and dropped to only 25% after 12 h. This result indicated that the excellent stability of 7.50% MnO₂@GR was attributed to the encapsulated structure.

Q6 In mean the possibility that ozone diffuse into the interlayer of graphene sheets (0.335 nm) rather than through benzene ring?

Response: We appreciate the valuable comments of Reviewer.

In the latest revision, we have already analyzed the two approaches for ozone molecule entering the interlayer of graphene sheets. On the one hand, ozone molecule cannot go through the hole of benzene ring since the size of

the ozone molecule (1.99Å) is much larger than that of benzene ring (1.33Å) as shown in Figure R2. On the other hand, ozone molecule diffusing into the interlayer of graphene sheets is theoretically possible but very slow, which will take longer than 150 s (Angew. Chem. Int. Ed. 2012, 124, 4940-4943).

In our catalyst system, the residence time of ozone molecule in catalyst bed is less than 1 s. Therefore, the possibility of the ozone diffusion into the interlayer of graphene sheets can be negligible in this specific system.

The details of calculating the residence time of ozone molecule in our specific catalyst bed:

The catalytic performance for ozone removal is evaluated in a fixed bed continuous flow quartz reactor (11 mm i.d.) at room temperature. 100 mg catalyst is filled in the quartz tube and the gas flowrate into the reactor is maintained at 900 mL/min (15 mL/s). After 7.50% MnO₂@GR is filled into the quartz tube, the height of the catalyst bed reaches 3 cm and hence the volume of the catalyst bed is 2.85 cm³. Therefore, the residence time of the airflow in the catalyst bed is only 0.19 s.

Figure R2. The model of graphene unit and ozone molecule.

Q7 Since the adsorption sites of water was difficult to be identified, what is the effect of water molecule on the as-proposed active site?

Response: We appreciate the valuable comments of Reviewer.

The experimental results of ozone decomposition on 7.50% MnO₂@GR under the condition of different relative humidity (20% RH and 50% RH) indicate that the water molecules enable to have competitive adsorption with the ozone molecules, which inhibit the catalytic activity and stability of 7.50% MnO₂@GR in the high humidity. But the adsorption strength of water molecules on the synthesized 7.50% MnO₂@GR is very weak as proved by our H₂O-TPD data (Figure R3, Figure 5a in the main manuscript), which only physically adsorbs on the synthesized catalyst. As a result, the deactivated catalyst induced by water can be easily re-generated by a drying process (110°C, air atmosphere).

As shown in figure 4c, when the humidity increases from 20% to 50%, the ozone conversion efficiency decreases rapidly, suggesting that the water still adsorbed on the catalyst surface. However, the adsorbed water molecules would not affect further ozone conversion under low humidity conditions, indicating that the sites for water adsorption and ozone conversion are different. The high concentration of water vapour, which results from the adsorption-enrichment of the surface hydrophilic groups (such as hydroxyl groups, C=O groups and COOH

groups), exacerbates the surface competitive adsorption, consequently resulting in water-induced deactivation.

Figure R3 TPD-MS profiles of MnO_2 and $7.50\% \text{ MnO}_2@GR$. Insets: Surface electrostatic potential and molecular dipole of O_3 and H_2O .

Q8 The authors may mean the surface hydroxyl groups ($-\text{OH}$) rather than surface hydroxide radical ($\bullet\text{OH}$) for the instability of the later one on metal or metal oxide surface. Additionally, there was no effective way to detect surface hydroxide radical.

Response: We appreciate the valuable comments of Reviewer.

We apologize for the mistake. The mentioned surface hydrophilic groups should be hydroxyl groups rather than surface hydroxide radicals. We have corrected the mistakes.

Reviewers' Comments:

Reviewer #1:

Remarks to the Author:

The manuscript has been improved a lot and can be published as the following issues are addressed:

(1) The uniformity of the graphene layer-encapsulated MnO₂ is very important. To the best of my knowledge, graphene layers-encapsulated metal nanoparticles such as Co, Ni, and Fe which are very sensitive to oxygen, can prevent the metallic species from being oxidized even after prolonged exposure to air (*Angew. Chem. Int. Ed.*, 2012, 52, 371-375; *Energy Environ. Sci.*, 2014, 7, 1919-1923; *Energy Environ. Sci.*, 2016, 9, 123-129). Goodenough et al. have detected the oxidized Fe, Co, and Ni of graphitic-shell encapsulated metal nanocatalysts after OER, ORR, and HER. This was only attributed to the access the metallic surface by electrolyte through the defect sites or removed amorphous carbon (*Adv. Energy Mater.* 2020, 10, 1903215.). Therefore, the rise in the oxidation state of MnO₂ in this work possibly happens by oxidation of exposed MnO₂ that was not encapsulated by graphene, which also contributed to the catalytic performance. The explanation on the oxidation of MnO₂ may be revised.

(2) The authors were right that XPS data and CV curves have confirmed the existence of the unsaturated Mn atoms. The active sites have been obtained based on the DFT calculations, in which the model were constructed using TEM, XPS and so on. However, the coordination environment of Mn that was very important for modeling was still missing. I insist that the coordination environment of Mn is very important for determining the active sites rather than only gaining fundamental insights in active sites from DFT calculations which was modeled based on the other techniques such as TEM, XPS, and UPS that provide general properties rather than local ones.

(3) You should check the cited Ref. (*Angew. Chem. Int. Ed.* 2012, 124, 4940-4943) in Q6. Please provide a right one.

Reviewer #3:

Remarks to the Author:

After several iterant comments and revisions, the authors greatly improved the quality of this manuscript according to the reviewers' suggestions, so this work could be accepted to be published in *Nature Communications* after carefully considering the forms thorough the manuscript including the form of the references.

1. According to the commonly accepted mechanism of the ozone decomposition on manganese oxides (*J. Am. Chem. Soc.* 1998, 120, 9041-9046; *J. Am. Chem. Soc.* 1998, 120, 9047-9052), the rate-limiting step is often considered to be the decomposition of adsorbed peroxide (O₂²⁻) intermediate to produce O₂ and the oxygen vacancy (Vo), i.e., Mn⁴⁺-O₂²⁻ → Mn²⁺-Vo + O₂ (or O₂²⁻ → O₂ + 2e⁻) (1). Thus, the stability of surface Mn²⁺ ions is critical for the rate of the ozone decomposition because it is extremely easy for reduced Mn²⁺ ions to be re-oxidized by O₃ to Mn⁴⁺ ions under the reaction conditions. As a consequence, in order to stabilize reduced Mn ions to allow them to be at a relatively steady state (*Ind. Eng. Chem. Res.* 2018, 57, 12590-12594), one strategy is to redistribute the charge of the reduced Mn ions to neighboring Mn ions (even on entire catalyst particles), or to other metallic/conductive materials such as graphene in this manuscript, as much as possible. One typical example is a previous report by the authors themselves (*Environ. Sci. Technol.* 2018, 52, 8684-8692), doping K⁺ in the tunnels of α-MnO₂ can enhance the conductivity of K-doped α-MnO₂ (*Nat. Commun.* 2016, 7:13374), thus improving the rate of the O₃ decomposition. This is the very reason that graphene can enhance the rate of the ozone decomposition on MnO₂, where graphene appears to act as a buffer pool or electrochemical capacitor (*Nano Lett.* 2011, 11, 2905-2911) for chemically-generated electrons during the reaction process so as to lower the activation energy especially for the elementary step (1). For this manuscript, even if there are some MnO₂ nanofibers that are not encapsulated by graphene, more actually exposed or not contacted with graphene, these MnO₂ nanofibers often show lower activity for the O₃ decomposition than graphene-'encapsulated' MnO₂

nanofibers. Hence, the conclusions the authors made should be relatively safe.

2. As for the identification of active sites, it is an extremely formidable task for scientists all over the world working in heterogeneous catalysis, even with assist of sophisticated instruments and advanced chemical molecular probes. I agree with the point commented by the reviewer 1# that the coordination environment of Mn is very important for identifying the active sites, and there is a gap between the statistical and average information of samples with nonuniform active sites from TEM, XPS, and UPS, and the simulations by DFT. In principle, the precise determination in structure of the active sites comes before the constructed modeling by DFT, but the surfaces of MnO₂ nanofibers with active sites possibly situated on the top-facets, side-facets and/or the edges (Chem. Eur. J. 2015, 21, 9619-9623) are very complex, which even become more complicated after encapsulation with graphene. As a result, this complex catalyst system makes the precise identification of active sites greatly challenging. During the reaction process, the coordination environment of active metal ions also often changes with reaction conditions because partially of the generation of V_O or its replenishment by oxygen atoms from O₃. As such, even to add the extra experiments, it is still very difficult to precisely identify active sites.

3. It is necessary for the authors to make a reasonable explanation for how to identify the active sites according to the suggestions raised by the reviewer, describing difficulties to do so due to many uncertainties. Probably, the active sites are situated on the MnO₂ surfaces, and graphene functions as an electron promoter as aforementioned. Because MnO₂ nanofibers often grow with the morphology of tetragonal prisms (J. Phys. Chem. B 2006, 110, 3066–3070), after the coverage of graphene as a curved-shaped tube even intimately contacted with the MnO₂ surfaces, there are still spaces enough for O₃ or O₂ molecules with the small sizes to gain access to active sites on MnO₂ surfaces to finish the reaction. Otherwise, if the defects or the functional group such as C-OH or C=O on graphene were defined as the active sites, it possibly occurs as a graphene-consuming chemical reaction not as a catalytic reaction; if active sites are localized on the graphene surface just over the coordinatively unsaturated Mn ions, as claimed the authors, sites for binding the active oxygen atom produced from decomposing the first O₃ molecule ($O_3 \rightarrow O_2 + O$) is also uncertain on graphene, and it is not feasible for this active oxygen atom to penetrate through the small-size benzene ring structure of graphene to bind to the coordinatively unsaturated Mn ions. Overall, from the application viewpoint, the merits of this manuscript more than offset the defect about identification of active sites, so I still recommend it for publication in Nature Communications.

Responses to the Reviewers

Reviewer #1:

Comments: The manuscript has been improved a lot and can be published as the following issues are addressed:

Response: We gratefully appreciate the suggestion of reviewer to publish our manuscript. Our manuscript has been revised according to the valuable suggestion of reviewer. All parts revised according to the reviewer's comments are shown in the revised manuscript and supplemental information files.

Q1 The uniformity of the graphene layer-encapsulated MnO₂ is very important. To the best of my knowledge, graphene layers-encapsulated metal nanoparticles such as Co, Ni, and Fe which are very sensitive to oxygen, can prevent the metallic species from being oxidized even after prolonged exposure to air (Angew. Chem. Int. Ed., 2012, 52, 371-375; Energy Environ. Sci., 2014, 7, 1919-1923; Energy Environ. Sci., 2016, 9, 123-129). Goodenough et al. have detected the oxidized Fe, Co, and Ni of graphitic-shell encapsulated metal nanocatalysts after OER, ORR, and HER. This was only attributed to the access the metallic surface by electrolyte through the defect sites or removed amorphous carbon (Adv. Energy Mater. 2020, 10, 1903215.). Therefore, the rise in the oxidation state of MnO₂ in this work possibly happens by oxidation of exposed MnO₂ that was not encapsulated by graphene, which also contributed to the catalytic performance. The explanation on the oxidation of MnO₂ may be revised.

Response: We appreciate the valuable comments of Reviewer.

We agree with the comments of Reviewer and the uniformity of the graphene layer is significant to understand the active site. So, we also try to investigate whether MnO₂ is exposed and the ratio of exposed MnO₂. However, the existing characterization technologies we can reach are hard to prove whether part of MnO₂ is encapsulated by graphene or not. On the other hand, our experimental results indicate that the MnO₂ nanofibers show much lower catalytic activity for the O₃ decomposition than that of graphene encapsulated MnO₂ nanofibers. Hence, even if there are some MnO₂ nanofibers that are not encapsulated by graphene, the contribution of these unencapsulated MnO₂ should make very limited contribution to the total catalytic performance. Therefore, it is clear that the excellent performance is attributed to the encapsulated structure even if there are some MnO₂ nanofibers that are not encapsulated by graphene.

As shown in figure 5d, the Mn 3s spectra of 7.5% MnO₂@GR showed that the oxidation states of MnO₂ are changed after reaction. On the one hand, the formation of C=O groups or COOH groups (figure 5c) would induce a stronger Mn-C bond, consequently resulting in a rise of the AOS of Mn. On the other hand, if MnO₂ is exposed, MnO₂ also can be oxidized by ozone molecule. Therefore, the explanation on the rise of the AOS of Mn was revised to make this point much clearer and comprehensive. The changes can be found in page 17, line 298-299.

Q2 The authors were right that XPS data and CV curves have confirmed the existence of the unsaturated Mn atoms. The active sites have been obtained based on the DFT calculations, in which the model were constructed using TEM, XPS and so on. However, the coordination environment of Mn that was very important for modeling was still missing. I insist that the coordination environment of Mn is very important for determining the active sites rather than only gaining fundamental insights in active sites from DFT calculations which was modeled based on the other techniques such as TEM, XPS, and UPS that provide general properties rather than local ones.

Response: We appreciate the valuable questions of Reviewer.

We agree with the comments of Reviewer and the coordination environment of Mn is an important factor in better understanding the active sites. However, identifying the active sites of oxides is an extremely formidable task for researchers in the community of heterogeneous catalysis. Especially, the surfaces of MnO₂ nanofibers are very complicated and the top-facets, side-facets and/or the edges possibly act as the active sites (Chem. Eur. J. 2015, 21, 9619-9623), which becomes even more complicated after encapsulation with graphene. Moreover, the dynamic changes on the composition, geometric structure and morphology of oxides under the reaction conditions make the identification on the active sites more complicated. To be honest, it is very difficult for us to precisely identify the active sites by using the characterization technologies we can reach. Hence, a DFT simulation was used trying our best to understand the catalytic active sites of graphene encapsulated α -MnO₂ nanofiber.

The detail process to identify the active sites was shown as following. Firstly, HRTEM and optical micrographs of MnO₂@GR confirmed the structure of graphene encapsulated α -MnO₂ nanofiber. The two reference hybrid catalysts (GO/MnO₂ and GO+MnO₂) indicated that the stability of MnO₂@GR was attributed to the encapsulated structure. For GO/MnO₂, α -MnO₂ nanowires were located on the surface of the large-areas graphene, while displays almost the same ozone conversion as that of α -MnO₂ nanowires, suggesting the stability of MnO₂@GR was not attributed to electron buffer of graphene layer. Secondly, the literature (J. Phys. Chem. C 2009, 113, 14225-14229) pointed out that oxygen species (O²⁻) formed on the pure graphene layer by dissociative chemisorption. However, the formed oxygen species (O²⁻) would not further react with ozone molecule, suggesting its lower electron density limited the electron donation for further ozone decomposition. Therefore, it can be concluded that the surface graphene was activated by inner α -MnO₂ nanofiber for ozone catalytic decomposition. Thirdly, XPS data and CV curves have confirmed the existence of the unsaturated Mn atoms. Therefore, electron transfer may appear in Mn-O-C and Mn-C simultaneously. So, DFT calculation was adopted to understand the interfacial electron transfer. The results indicated that the electron transfer from graphite carbon to oxygen atoms, reducing the surface electron density and not beneficial for ozone catalytic decomposition, while the electron transfer from the unsaturated Mn atoms to graphite carbon, increasing the surface electron density and beneficial for ozone catalytic decomposition. Therefore, the graphite carbon near unsaturated Mn atoms was the active sites

for ozone catalytic decomposition in MnO₂@GR.

Q3 You should check the cited Ref. (Angew. Chem. Int. Ed. 2012, 124, 4940-4943) in Q6. Please provide a right one.

Response: we appreciate the suggestion of the Reviewer.

According to the suggesting, we checked the reference and corrected “Angew. Chem. Int. Ed. 2012, 124, 4940-4943” as “Angew. Chem. Int. Ed. 2012, 51, 4856-4859”. This reference pointed out gas molecule diffusing into the interlayer of graphene sheets is theoretically possible but very slow. Therefore, we propose that the graphitic carbon close to Mn atoms (oxygen vacancy) should be the active site for ozone decomposition in MnO₂@GR.

Reviewer #3:

Comments: After several iterant comments and revisions, the authors greatly improved the quality of this manuscript according to the reviewers’ suggestions, so this work could be accepted to be published in Nature Communications after carefully considering the forms thorough the manuscript including the form of the references.

We gratefully appreciate the suggestion of reviewer to publish our manuscript.

Q1 According to the commonly accepted mechanism of the ozone decomposition on manganese oxides (J. Am. Chem. Soc. 1998, 120, 9041–9046; J. Am. Chem. Soc. 1998, 120, 9047–9052), the rate-limiting step is often considered to be the decomposition of adsorbed peroxide (O₂²⁻) intermediate to produce O₂ and the oxygen vacancy (Vo), i.e., Mn⁴⁺-O₂²⁻ → Mn²⁺-Vo + O₂ (or O₂²⁻ → O₂ + 2e-) (1). Thus, the stability of surface Mn²⁺ ions is critical for the rate of the ozone decomposition because it is extremely easy for reduced Mn²⁺ ions to be re-oxidized by O₃ to Mn⁴⁺ ions under the reaction conditions. As a consequence, in order to stabilize reduced Mn ions to allow them to be at a relatively steady state (Ind. Eng. Chem. Res. 2018, 57, 12590-12594), one strategy is to redistribute the charge of the reduced Mn ions to neighboring Mn ions (even on entire catalyst particles), or to other metallic/conductive materials such as graphene in this manuscript, as much as possible. One typical example is a previous report by the authors themselves (Environ. Sci. Technol. 2018, 52, 8684–8692), doping K⁺ in the tunnels of α-MnO₂ can enhance the conductivity of K-doped α-MnO₂ (Nat. Commun. 2016, 7:13374), thus improving the rate of the O₃ decomposition. This is the very reason that graphene can enhance the rate of the ozone decomposition on MnO₂, where graphene appears to act as a buffer pool or electrochemical capacitor (Nano Lett. 2011, 11, 2905–2911) for chemically-generated electrons during the reaction process so as to lower the activation energy especially for the elementary step (1). For this manuscript, even if there are some MnO₂ nanofibers that are

not encapsulated by graphene, more actually exposed or not contacted with graphene, these MnO₂ nanofibers often show lower activity for the O₃ decomposition than graphene-‘encapsulated’ MnO₂ nanofibers. Hence, the conclusions the authors made should be relatively safe.

Response: we appreciate the Reviewer’s agreement on the strategy of encapsulating MnO₂ by graphene we proposed in this manuscript to enhance the catalytic performance of ozone decomposition.

The two reference hybrid catalysts (GO/MnO₂ and GO+MnO₂) indicated that the stability of MnO₂@GR was attributed to the encapsulated structure. For GO/MnO₂, α-MnO₂ nanowires were located on the surface of the large-areas graphene, while displays almost the same ozone conversion as that of α-MnO₂ nanowires, suggesting the stability of MnO₂@GR was not attributed to electron buffer of graphene layer. In MnO₂@GR, electron transfer may appear in Mn-O-C and Mn-C simultaneously. So, DFT calculation was adopted to understand the interfacial electron transfer. The results indicated that the electron transfer from graphite carbon to oxygen atoms, reducing the surface electron density and not beneficial for ozone catalytic decomposition, while the electron transfer from the unsaturated Mn atoms to graphite carbon, increasing the surface electron density and beneficial for ozone catalytic decomposition. Therefore, the graphite carbon near unsaturated Mn atoms was the active sites for ozone catalytic decomposition in MnO₂@GR.

Q2 As for the identification of active sites, it is an extremely formidable task for scientists all over the world working in heterogeneous catalysis, even with assist of sophisticated instruments and advanced chemical molecular probes. I agree with the point commented by the reviewer 1# that the coordination environment of Mn is very important for identifying the active sites, and there is a gap between the statistical and average information of samples with nonuniform active sites from TEM, XPS, and UPS, and the simulations by DFT. In principle, the precise determination in structure of the active sites comes before the constructed modeling by DFT, but the surfaces of MnO₂ nanofibers with active sites possibly situated on the top-facets, side-facets and/or the edges (Chem. Eur. J. 2015, 21, 9619-9623) are very complex, which even become more complicated after encapsulation with graphene. As a result, this complex catalyst system makes the precise identification of active sites greatly challenging. During the reaction process, the coordination environment of active metal ions also often changes with reaction conditions because partially of the generation of V0 or its replenishment by oxygen atoms from O₃. As such, even to add the extra experiments, it is still very difficult to precisely identify active sites.

Response: we completely agree with the comments of Reviewers. It is true that the identification on the active sites of oxides like MnO₂ is extremely challenging since the top-facets, side-facets and/or the edges of oxides enable to act as active sites, which makes the composition of active sites very complicated.

Q3 It is necessary for the authors to make a reasonable explanation for how to identify the active sites according to

the suggestions raised by the reviewer, describing difficulties to do so due to many uncertainties. Probably, the active sites are situated on the MnO₂ surfaces, and graphene functions as an electron promoter as aforementioned. Because MnO₂ nanofibers often grow with the morphology of tetragonal prisms (J. Phys. Chem. B 2006, 110, 3066–3070), after the coverage of graphene as a curved-shaped tube even intimately contacted with the MnO₂ surfaces, there are still spaces enough for O₃ or O₂ molecules with the small sizes to gain access to active sites on MnO₂ surfaces to finish the reaction. Otherwise, if the defects or the functional group such as C-OH or C=O on graphene were defined as the active sites, it possibly occurs as a graphene-consuming chemical reaction not as a catalytic reaction; if active sites are localized on the graphene surface just over the coordinatively unsaturated Mn ions, as claimed the authors, sites for binding the active oxygen atom produced from decomposing the first O₃ molecule ($O_3 \rightarrow O_2 + O$) is also uncertain on graphene, and it is not feasible for this active oxygen atom to penetrate through the small-size benzene ring structure of graphene to bind to the coordinatively unsaturated Mn ions. Overall, from the application viewpoint, the merits of this manuscript more than offset the defect about identification of active sites, so I still recommend it for publication in Nature Communications.

Response: we appreciate the Reviewer's recommendation to publish our work. According your suggestions, we explained the detail process to identify the active sites.